# Statistical Inference for Deep Learning via Stochastic Modeling

## Abstract

Deep learning has revolutionized big data analysis in modern data science, however, how to make statistical inference for deep neural networks remains largely unclear. To this end, we explore a stochastic variant of the deep neural network known as the stochastic neural network (StoNet), as developed in Liang et al. (2022). Firstly, we show that the StoNet falls into the framework of statistical modeling. It not only enables us to address fundamental issues in deep learning, such as structure interpretability and uncertainty quantification, but also provides with us a platform for transferring the theory and methods developed for linear models to the realm of deep learning. Specifically, we show how the sparse learning theory with the Lasso penalty can be adapted to deep neural networks (DNNs) from linear models; establish that the sparse StoNet is consistent in network structure selection; and provides a recursive method to quantify the prediction uncertainty for the StoNet. Furthermore, we extend this result to the DNN by its asymptotic equivalence with the StoNet, showing that consistent sparse deep learning can be obtained by training a DNN with an appropriate Lasso penalty. Additionally, we propose to remodel the last hidden layer output and the target output of a well trained DNN model using a StoNet on the validation dataset, and then assess the prediction uncertainty of the DNN model via the StoNet. The proposed method has been compared with conformal inference on extensive examples, and numerical results suggests its superiority.

## 1 Introduction

Over the past two decades, deep learning has revolutionized modern data science, achieving remarkable success in many scientific fields, such as pattern recognition, protein structure prediction, and natural language processing. However, from the perspective of statistical modeling, the deep neural network (DNN) still suffers from a fundamental issue: overparameterization. Consequently, the training data is often overfitted and the downstream statistical inference cannot be effectively conducted. In particular, the structure of the DNN is difficult to interpret, and its prediction uncertainty is challenging to quantify (Guo et al.,

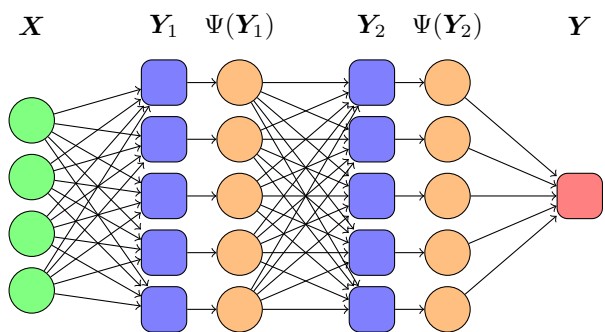

Figure 1: Illustration of the structure of the StoNet, where each square neuron represents a linear/logistic regression: $\boldsymbol{X}$ represents input variable, $\boldsymbol{Y}_1 = \boldsymbol{b}_1 + \boldsymbol{w}_1 X + \boldsymbol{e}_1$ and $\boldsymbol{Y}_2 = \boldsymbol{b}_2 + \boldsymbol{w}_2 \Psi(\boldsymbol{Y}_1) + \boldsymbol{e}_2$ represent latent variables, $\boldsymbol{Y} = \boldsymbol{b}_3 + \boldsymbol{w}_3 \Psi(\boldsymbol{Y}_2) + \boldsymbol{e}_3$ represent output variables, and $\Psi(\cdot)$ represents the activation function.

2017). On the other hand, statistics, in its century-long history, has developed principles and methods to tackle issues like overparametrization and uncertainty quantification for traditional statistical models, especially linear models. This naturally raises a question: Can

we bridge the gap between linear models and DNNs, enabling the adaptation of theory and methods from linear models to DNNs to effectively address these issues?

We find that this question can be satisfactorily addressed with a new type of stochastic neural network (StoNet), which is formulated as a composition of many simple linear/logistic regressions. The StoNet is asymptotically equivalent to the DNN in function approximation, but its structure is more interpretable from statistical perspectives and more designable in serving different structures of data. Figure 1 depicts the structure of a StoNet, where the complex deep learning task is broken down into many simple, neuron-wise linear/logistic regressions by adding random errors to the feeding values of each hidden neuron (see Section 2 for its mathematical formulas). The logistic regression is used only at the output layer for classification problems. The StoNet was first proposed by the authors in Liang et al. (2022) and Sun & Liang (2022b), but its property as a bridge between linear models and deep learning is first explored in this paper. In Liang et al. (2022), it is shown that the fully connected StoNet can be used as a general tool of nonlinear sufficient dimension reduction for large-scale data. In Sun & Liang (2022b), the StoNet works with a restrictive structure, where each linear regression at the first hidden layer is replaced by a support vector regression (SVR), and as a result, it avoids the local trap issue in training.

**Our Contributions**   This paper conducts a full exploration of the properties of the StoNet as a bridge between linear models and deep learning, and it provides solid solutions for addressing the issues such as overparametrization, structural interpretability, and prediction uncertainty quantification. More precisely,

- We show that the sparse learning theory and methods developed for linear models, such as those utilizing Lasso (Tibshirani, 1996) and other amenable penalties (Loh & Wainwright, 2017), can be effectively adapted to the StoNet. The resulting sparse StoNet fits into the framework of statistical modeling: it exhibits consistency in structure selection even when the sizes of the input and hidden layers are much larger than the training sample size. Furthermore, its prediction uncertainty can be quantified through a recursive application of Eve's law.

- By the asymptotic equivalence between the StoNet and DNN, we justify the consistency of sparse deep learning with the Lasso penalty. While the approach has long been commonly practiced in the community, see e.g. Scardapane et al. (2017) and Lemhadri et al. (2019), to the best of our knowledge, the consistency theory supporting this practice has not been previously established.

- Based on certain theoretical properties of the DNN and StoNet, as well as the feasibility of applying the sparse StoNet for uncertainty quantification, we propose a post-StoNet procedure to quantify the prediction uncertainty of large-scale DNNs. Numerical comparisons with the conformal method (Vovk et al., 2005; Shafer & Vovk, 2008) indicate the superiority of the post-StoNet procedure.

**Related Works**   Stochastic neural networks have a long history in machine learning, with famous examples including deep belief networks (Hinton & Salakhutdinov, 2006; Hinton, 2007) and deep Boltzmann machines (Salakhutdinov & Hinton, 2009). In recent years, researchers have proposed various techniques to introduce noise into DNNs in order to enhance their performance. For instance, Srivastava et al. (2014) introduced the dropout method, which randomly drops hidden and visible neurons during training to prevent overfitting. Neelakantan et al. (2017) suggested adding gradient noise to improve training. Other approaches involve using noisy activation functions to improve generalization and adversarial robustness (Gülçehre et al., 2016; Noh et al., 2017; You et al., 2018), as well as learning uncertainty parameters of stochastic activation functions alongside neural network training (Yu et al., 2021). However, it is not clear if these models provide valid probabilistic approximation for conventional DNNs. Moreover, how to provide valid uncertainty quantification based on these models is not well studied.

From the perspective of bridging linear models and DNNs, there is a related branch of work focusing on neural tangent kernel (Jacot et al., 2018). Specifically, Liu et al. (2020) and Liu et al. (2021) explore the transition to linearity of the DNN model and, equivalently, constancy of the neural tangent kernel, based on the scaling properties of its Hessian matrix

as a function of the network width. However, their exploration primarily focuses on the loss landscapes, which is quite distinct from our general goal of statistical inference.

## 2 StoNet and Its Approximation to DNN

Consider a DNN model with $h$ hidden layers. For the sake of simplicity, we assume that the same activation function $\psi(\cdot)$ is used for all hidden neurons. By separating the feeding and activation operators of each hidden neuron, we can rewrite the DNN in the form:

$$\tilde{\boldsymbol{Y}}_1 = \boldsymbol{b}_1 + \boldsymbol{w}_1\boldsymbol{X}, \quad \tilde{\boldsymbol{Y}}_i = \boldsymbol{b}_i + \boldsymbol{w}_i\Psi(\tilde{\boldsymbol{Y}}_{i-1}), \quad i = 2, 3, \ldots, h,$$
$$\boldsymbol{Y} = \boldsymbol{b}_{h+1} + \boldsymbol{w}_{h+1}\Psi(\tilde{\boldsymbol{Y}}_h) + \boldsymbol{e}_{h+1}, \tag{1}$$

where $\boldsymbol{e}_{h+1} \sim N(0, \sigma_{h+1}^2 I_{d_{h+1}})$ is Gaussian random error; $\tilde{\boldsymbol{Y}}_i, \boldsymbol{b}_i \in \mathbb{R}^{d_i}$ for $i = 1, 2, \ldots, h$; $\boldsymbol{Y}, \boldsymbol{b}_{h+1} \in \mathbb{R}^{d_{h+1}}$; $\Psi(\tilde{\boldsymbol{Y}}_{i-1}) = (\psi(\tilde{Y}_{i-1,1}), \psi(\tilde{Y}_{i-1,2}), \ldots, \psi(\tilde{Y}_{i-1,d_{i-1}}))^T$ for $i = 2, 3, \ldots, h+1$, $\psi(\cdot)$ is the activation function, and $\tilde{Y}_{i-1,j}$ is the $j$th element of $\tilde{\boldsymbol{Y}}_{i-1}$; $\boldsymbol{w}_i \in \mathbb{R}^{d_i \times d_{i-1}}$ for $i = 1, 2, \ldots, h+1$, and $d_0 = p$ denotes the dimension of $\boldsymbol{X}$. For simplicity, we consider only regression problems in (1). By replacing the third equation of (1) with a logit model, the DNN can be extended to classification problems.

The StoNet, as illustrated by Figure 1, is *a probabilistic deep learning model* and constructed by adding auxiliary noise to $\tilde{\boldsymbol{Y}}_i$'s in (1). Mathematically, the StoNet is given by

$$\boldsymbol{Y}_1 = \boldsymbol{b}_1 + \boldsymbol{w}_1\boldsymbol{X} + \boldsymbol{e}_1, \quad \boldsymbol{Y}_i = \boldsymbol{b}_i + \boldsymbol{w}_i\Psi(\boldsymbol{Y}_{i-1}) + \boldsymbol{e}_i, \quad i = 2, 3, \ldots, h,$$
$$\boldsymbol{Y} = \boldsymbol{b}_{h+1} + \boldsymbol{w}_{h+1}\Psi(\boldsymbol{Y}_h) + \boldsymbol{e}_{h+1}, \tag{2}$$

as a composition of many simple regressions, where $\boldsymbol{Y}_1, \boldsymbol{Y}_2, \ldots, \boldsymbol{Y}_h$ are latent variables. For simplicity, we assume that $\boldsymbol{e}_i \sim N(0, \sigma_i^2 I_{d_i})$ for $i = 1, 2, \ldots, h, h+1$. However, other distributions can also be assumed for $\boldsymbol{e}_i$'s. For instance, Sun & Liang (2022a) assumed a modified double exponential distribution for $\boldsymbol{e}_1$ such that support vector regression applies. For classification problems, $\sigma_{h+1}^2$ plays the role of temperature for the binomial or multinomial distribution formed at the output layer, and it works with $\{\sigma_1^2, \ldots, \sigma_h^2\}$ together to control the variation of the latent variables $\{\boldsymbol{Y}_1, \ldots, \boldsymbol{Y}_h\}$. For regression problems, this is similar.

The property of the StoNet as an approximator to the DNN, has been studied in Liang et al. (2022). A brief review for their theory is provided as follows, which form the basis for this work. Let $\boldsymbol{\theta} = (\boldsymbol{w}_1, \boldsymbol{b}_1, \ldots, \boldsymbol{w}_{h+1}, \boldsymbol{b}_{h+1})$ denote the collection of all weights of the StoNet (2), let $\boldsymbol{\Theta}$ denote the space of $\boldsymbol{\theta}$, and let $\boldsymbol{Y}_{\text{mis}} = (\boldsymbol{Y}_1, \boldsymbol{Y}_2, \ldots, \boldsymbol{Y}_h)$ denote the collection of all latent variables. Let $\pi(\boldsymbol{Y}, \boldsymbol{Y}_{\text{mis}}|\boldsymbol{X}, \boldsymbol{\theta})$ denote the likelihood function of the StoNet (2), and let $\pi_{\text{DNN}}(\boldsymbol{Y}|\boldsymbol{X}, \boldsymbol{\theta})$ denote the likelihood function of the DNN model (1).

**Lemma 1.** *(Liang et al., 2022) Suppose that Assumptions A1-A2 hold, and $\pi(\boldsymbol{Y}, \boldsymbol{Y}_{\text{mis}}|\boldsymbol{X}, \boldsymbol{\theta})$ is continuous in $\boldsymbol{\theta}$. Then*

$$(i) \quad \sup_{\boldsymbol{\theta} \in \boldsymbol{\Theta}} \left| \frac{1}{n} \sum_{i=1}^n \log \pi(\boldsymbol{Y}^{(i)}, \boldsymbol{Y}_{mis}^{(i)}|\boldsymbol{X}^{(i)}, \boldsymbol{\theta}) - \frac{1}{n} \sum_{i=1}^n \log \pi_{\text{DNN}}(\boldsymbol{Y}^{(i)}|\boldsymbol{X}^{(i)}, \boldsymbol{\theta}) \right| \xrightarrow{p} 0, \tag{3}$$

$$(ii) \quad \|\hat{\boldsymbol{\theta}}_n - \boldsymbol{\theta}^*\| \xrightarrow{p} 0, \quad as\ n \to \infty,$$

*where $\boldsymbol{\theta}^* = \arg\max_{\boldsymbol{\theta} \in \boldsymbol{\Theta}} \mathbb{E}(\log \pi_{\text{DNN}}(\boldsymbol{Y}|\boldsymbol{X}, \boldsymbol{\theta}))$ denotes the true parameters of the DNN model as specified in (1), and $\hat{\boldsymbol{\theta}}_n = \arg\max_{\boldsymbol{\theta} \in \boldsymbol{\Theta}}\{\frac{1}{n}\sum_{i=1}^n \log \pi(\boldsymbol{Y}^{(i)}, \boldsymbol{Y}_{mis}^{(i)}|\boldsymbol{X}^{(i)}, \boldsymbol{\theta})\}$ denotes the maximum likelihood estimator of the StoNet model (2) with the pseudo-complete data.*

Lemma 1 implies that the StoNet and DNN are asymptotically equivalent as the training sample size $n$ becomes large, and it forms the basis for the bridging property of the StoNet. The asymptotic equivalence can be elaborated from two perspectives. First, suppose the DNN model (1) is true. Lemma 1 implies that when $n$ becomes large, the weights of the DNN can be learned by training a StoNet of the same structure with $\sigma_i^2$'s satisfying Assumption A1-(v). *Algorithmically, the StoNet provides an alternative way to train the DNN using latent variable augmentation when the training sample size is large.* On the other hand, suppose the StoNet (2) is true. Then Lemma 1 implies that *for any StoNet satisfying Assumptions A1 & A2, the weights $\boldsymbol{\theta}$ can be learned by training a DNN of the same structure when the*

*training sample size is large.* As shown later, this asymptotic equivalence leads to interesting results toward post-training inference for large-scale DNNs.

Lemma 1 also implies that, like the DNN, the StoNet possesses the universal approximation property for representing probability distributions. Refer to Lu & Lu (2020) for the establishment of this property for the DNN. Theoretically, all the DNN approximation properties can be carried over to the StoNet.

## 3 Sparse StoNet Learning

This section describes two algorithms for sparse StoNet learning. One is based on the imputation-regularized optimization (IRO) algorithm (Liang et al., 2018), and the other is based on an adaptive stochastic gradient MCMC(ASGMCMC) algorithm (Liang et al., 2022). Details of the algorithms are given in section B in appendix. The IRO algorithm is a stochastic EM type algorithm, which provides us with a framework to transfer the theory and methods from linear models to deep learning, while the ASGMCMC algorithm allows us to train the StoNet more efficiently with the use of the mini-batching.

### 3.1 The IRO Algorithm and Consistency of Sparse StoNets

*Notations:* In this subsection, we will rewrite the network depth $h$ as $h_n$, rewrite the network widths $(p, d_1, \ldots, d_{h+1})$ as $(p_n, d_{1,n}, \ldots, d_{h+1,n})$, rewrite the layer-wise variance $\boldsymbol{\sigma}^2 = (\sigma_1^2, \ldots, \sigma_{h+1}^2)$ as $\boldsymbol{\sigma}_n^2 = (\sigma_{1,n}^2, \ldots, \sigma_{h+1,n}^2)$, where the subscript $n$ indicates their dependency on the training sample size. For simplicity of theoretical development, we will assume that for a given dataset $D_n = (\boldsymbol{Y}, \boldsymbol{X})$, the true model is a StoNet model with $\boldsymbol{\sigma}_n^2$ being known and satisfying Assumption A1-(v).

By treating the latent variables $\boldsymbol{Y}_{\text{mis}}$ as missing data, the IRO algorithm can be applied for training the StoNet. The IRO algorithm starts with an initial estimate of $\boldsymbol{\theta}$, denoted by $\hat{\boldsymbol{\theta}}_n^{(0)}$, and then iterates between the imputation and regularized optimization steps as shown in Algorithm 1. The key to the IRO algorithm is to find a sparse estimator for the working true parameter $\boldsymbol{\theta}_*^{(t)}$, as defined in equation (A3), that is uniformly consistent over all iterations. As suggested by Liang et al. (2018), such a uniformly consistent sparse estimator can typically be obtained by minimizing an appropriately penalized loss function as defined in (A2). For the StoNet, solving (A2) corresponds to solving a series of linear regressions by noting that the joint distribution $\pi(\boldsymbol{Y}_{\text{mis}}, \boldsymbol{Y}|\boldsymbol{X}, \boldsymbol{\theta}_n, \boldsymbol{\sigma}_n^2)$ can be decomposed in a Markov structure:

$$\pi(\boldsymbol{Y}_{\text{mis}}, \boldsymbol{Y}|\boldsymbol{X}, \boldsymbol{\theta}_n, \boldsymbol{\sigma}_n^2) = \pi(\boldsymbol{Y}|\boldsymbol{Y}_h, \boldsymbol{\theta}_n, \boldsymbol{\sigma}_n^2)\pi(\boldsymbol{Y}_h|\boldsymbol{Y}_{h-1}, \boldsymbol{\theta}_n, \boldsymbol{\sigma}_n^2)\cdots\pi(\boldsymbol{Y}_1|\boldsymbol{X}, \boldsymbol{\theta}_n, \boldsymbol{\sigma}_n^2), \quad (4)$$

and, furthermore, the components of $\boldsymbol{Y}_i \in \mathbb{R}^{d_i}$ are mutually independent conditional on $\boldsymbol{Y}_{i-1}$ for $i = 1, 2, \ldots, h+1$.

Suppose that the Lasso penalty (Tibshirani, 1996) is imposed on $\boldsymbol{\theta}$. Theorem 1 shows that the resulting IRO estimator $\hat{\boldsymbol{\theta}}_n^{(t)}$ is consistent when both $n$ and $t$ are sufficiently large, and the underlying true sparse StoNet can be consistently identified.

**Theorem 1.** *Suppose that the Lasso penalty is imposed on $\boldsymbol{\theta}$, and Assumptions A1-A4 hold.*

(i) *There exist some constants $c_1 > 0$, $c_2 > 0$ and $c_3 > 0$ such that $\|\hat{\boldsymbol{\theta}}_n^{(t)} - \boldsymbol{\theta}_*^{(t)}\|_2^2 \prec r_n = o(1)$ holds uniformly for all iterations, where*

$$r_n = c_1 \frac{\sigma_{1,n}^2}{\kappa_{\min}^2} d_{1,n} p_n^s \frac{\log p_n}{n} + c_2 \sum_{l=2}^{h+1} \frac{\sigma_{l,n}^2}{\sigma_{l-1,n}^4} d_{l,n} d_{l-1,n}^s \frac{\log d_{l-1,n}}{n}$$

*for the StoNet with a linear regression output layer,*

$$r_n = c_1 \frac{\sigma_{1,n}^2}{\kappa_{\min}^2} d_{1,n} p_n^s \frac{\log p_n}{n} + c_2 \sum_{l=2}^{h} \frac{\sigma_{l,n}^2}{\sigma_{l-1,n}^4} d_{l,n} d_{l-1,n}^s \frac{\log d_{l-1,n}}{n} + \frac{c_3}{\sigma_{h,n}^4} d_{h+1,n} d_{h,n}^s \frac{(\log d_{h,n})^{2/3}}{n^{2(1-\varepsilon)/3}}$$

*for the StoNet with the logistic regression output layer.*

*(ii) Furthermore, if Assumption A5 holds, then $\|\hat{\boldsymbol{\theta}}_n^{(t)} - \boldsymbol{\theta}^*\| \xrightarrow{p} 0$ for sufficiently large n and sufficiently large t and almost every training dataset $D_n$.*

*(iii) Additionally, if Assumption A6 also holds and we select the connections by setting $\widehat{\boldsymbol{\gamma}}_n^{(t)} := \{i : |\hat{\theta}_{i,n}^{(t)}| > c\sqrt{r_n}\}$ for some constant c, where $\hat{\theta}_{i,n}^{(t)}$ denotes the i-th component of $\hat{\boldsymbol{\theta}}_n^{(t)}$, then the selected model $\widehat{\boldsymbol{\gamma}}_n^{(t)}$ is a consistent estimator of the true model. That is, $P(\widehat{\boldsymbol{\gamma}}_n^{(t)} = \boldsymbol{\gamma}^*) \to 1$ as $n \to \infty$ and $t \to \infty$.*

In summary, we have given a constructive proof for the consistency of sparse StoNets based on the sparse learning theory developed for linear models. Our proof implies that under Assumptions A1-A6, such a consistent estimator can also be obtained by directly maximizing the penalized log-likelihood function of the complete data, i.e., setting

$$\widehat{\boldsymbol{\theta}}_n^* = \arg\max_{\boldsymbol{\theta}} \left\{ \frac{1}{n} \sum_{i=1}^n \log \pi(\boldsymbol{Y}^{(i)}, \boldsymbol{Y}_{\text{mis}}^{(i)} | \boldsymbol{X}^{(i)}, \boldsymbol{\theta}, \boldsymbol{\sigma}^2) + \frac{1}{n} P_\lambda(\boldsymbol{\theta}) \right\}, \tag{5}$$

where $P_\lambda(\boldsymbol{\theta})$ satisfies Assumption A4. Furthermore, it follows from Lemma 1 that a consistent estimator of $\boldsymbol{\theta}$ can also be obtained by directly maximizing the penalized log-likelihood function of the DNN model, i.e., setting

$$\widehat{\boldsymbol{\theta}}_{\text{DNN,n}}^* = \arg\max_{\boldsymbol{\theta}} \left\{ \frac{1}{n} \sum_{i=1}^n \log \pi(\boldsymbol{Y}^{(i)} | \boldsymbol{X}^{(i)}, \boldsymbol{\theta}) + \frac{1}{n} P_\lambda(\boldsymbol{\theta}) \right\}, \tag{6}$$

for the same penalty function $P_\lambda(\boldsymbol{\theta})$ as used in (5). In summary, we have the corollary:

**Corollary 1.** *If Assumptions A1-A6 hold and the Lasso penalty $P_\lambda(\boldsymbol{\theta})$ is employed, then the estimator (6) is consistent in both parameter estimation and structure selection, i.e., $\|\widehat{\boldsymbol{\theta}}_{\text{DNN,n}}^* - \boldsymbol{\theta}^*\| \xrightarrow{p} 0$ and $P(\widehat{\boldsymbol{\gamma}}_{\text{DNN,n}} = \boldsymbol{\gamma}^*) \to 1$ as $n \to \infty$, where $\widehat{\boldsymbol{\gamma}}_{\text{DNN,n}}$ is selected with the same threshold as given in part (iii) of Theorem 1.*

We note that training sparse DNNs with the Lasso penalty has long been commonly practiced in the community, see e.g. Scardapane et al. (2017) and Lemhadri et al. (2019). However, to the best of our knowledge, the consistency theory supporting this has not been previously established. Speifically, Corollary 1 establishes consistency for both DNN parameter estimation and structure selection. Although our results were stated for model 1 and 2 for simplicity, it can be extended to other neural network structure such as convolutional neural network, where pre-activation hidden neurons are obtained by linear operations of the output of previous layers.

**Remark 1.** *Since the true value of $\sigma_{h+1,n}^2$ is generally unknown for the model (2), setting $\sigma_{h+1,n}^2$ to a smaller value corresponds to overfitting the model in imputation. This mimics a small-n-large-p linear/logistic regression problem, for which the fitting error of the full model can be very close to 0 while the true model can still be correctly identified with an appropriate amenable penalty (Loh & Wainwright, 2017). In our numerical experiments, see Sections 5-7, we often set $\sigma_{h+1,n}^2$ and thus $\sigma_{l,n}^2$'s (for $l = 1, 2, \ldots, h_n$) very small values and then perform variable selection for associated regressions with the Lasso penalty.*

### 3.2 Adaptive Stochastic Gradient MCMC for Efficient StoNet Learning

The IRO algorithm is developed under the full data setting and thus less scalable with respect to big data. To address this issue, we can train the sparse StoNet using an adaptive stochastic gradient MCMC algorithm (Liang et al., 2022), which is designed to find the maximum *a posteriori* (MAP) estimator $\hat{\boldsymbol{\theta}}_n^* = \arg\max_{\boldsymbol{\theta}} \{\pi(\boldsymbol{Y} | \boldsymbol{X}, \boldsymbol{\theta}, \boldsymbol{\sigma}^2) P_\lambda(\boldsymbol{\theta})\}$ by solving the equation:

$$\int \nabla_{\boldsymbol{\theta}} [\log \pi(\boldsymbol{Y}, \boldsymbol{Y}_{\text{mis}} | \boldsymbol{X}, \boldsymbol{\theta}, \boldsymbol{\sigma}^2) + \log P_\lambda(\boldsymbol{\theta})] \pi(\boldsymbol{Y}_{\text{mis}} | \boldsymbol{Y}, \boldsymbol{X}, \boldsymbol{\theta}, \boldsymbol{\sigma}^2) d\boldsymbol{Y}_{\text{mis}} = 0,$$

where $P_\lambda(\boldsymbol{\theta})$ denotes a penalty function satisfying Assumption A4. This algorithm is scalable for big data by making use of mini-batch samples at each iteration. Since the algorithm is

a slight modification of the algorithm in Liang et al. (2022), we leave it to the Appendix to ensure the paper is self-contained. As shown in Lemma A2, this algorithm also yield a consistent estimate of $\boldsymbol{\theta}$. Consequently, under Assumption A6, the structure of the sparse StoNet can also be consistently identified.

## 4    Prediction Uncertainty Quantification for Sparse StoNet

The hierarchical structure of the StoNet enables us to quantify the uncertainty of the latent variable at each layer using Eve's law in a recursive way. Assume that the StoNet is trained by the IRO algorithm. Let $\boldsymbol{z}$ denote a test point at which the prediction uncertainty is to be quantified, let $\boldsymbol{Z}_i^{(t)}$ denote the latent variable at layer $i$, corresponding to the input vector $\boldsymbol{z}$, imputed based on the estimate of $\boldsymbol{\theta}$ at iteration $t$ of the algorithm. Let $\boldsymbol{\mu}_i^{(t)}$ and $\boldsymbol{\Sigma}_i^{(t)}$ denote, respectively, the mean and covariance matrix of $\boldsymbol{Z}_i^{(t)}$. By Eve's law, for any layer $i \in \{2, 3, \ldots, h+1\}$, we have $\boldsymbol{\Sigma}_i^{(t)} = \mathbb{E}(\mathrm{Var}(\boldsymbol{Z}_i^{(t)}|\boldsymbol{Z}_{i-1}^{(t)})) + \mathrm{Var}(\mathbb{E}(\boldsymbol{Z}_i^{(t)}|\boldsymbol{Z}_{i-1}^{(t)}))$. As detailed in Section E, the final formula (A12) can be derived. This leads to the following procedure for prediction interval construction.

Let $\mu(\boldsymbol{z}, \hat{\boldsymbol{\theta}})$ denote the prediction of a StoNet with weights $\hat{\boldsymbol{\theta}}$ at point $\boldsymbol{z}$. Note that the StoNet (2) has the same prediction function as the DNN (1), i.e., the random noise added to the latent variables is set to 0 in forward prediction. Suppose that a set of StoNet estimates, $\mathcal{S} = \{\hat{\boldsymbol{\theta}}^{(1)}, \hat{\boldsymbol{\theta}}^{(2)}, \ldots, \hat{\boldsymbol{\theta}}^{(m)}\}$, has been collected after convergence of the IRO algorithm. Given $\widehat{\Sigma}_i^{(t)}$'s, by the Wald method, the 95% prediction interval of $\mu_j(\boldsymbol{z}, \boldsymbol{\theta}^*)$, the $j$-th component of $\mu(\boldsymbol{z}, \boldsymbol{\theta}^*)$, can be constructed in the following procedure:

(i) For each StoNet estimate $\hat{\boldsymbol{\theta}}^{(t)} \in \mathcal{S}$, calculate the variance of the training error by $\hat{\varsigma}_{h+1,j}^{2(t)} = \frac{1}{n} \sum_{k=1}^n (\mu_j(\boldsymbol{x}^{(k)}, \hat{\boldsymbol{\theta}}^{(t)}) - y_j^{(k)})^2$, where $(\boldsymbol{x}^{(k)}, \boldsymbol{y}^{(k)})$ denotes the $k$-th training sample, and $y_j^{(k)}$ is the $j$-th component of $\boldsymbol{y}^{(k)}$.

(ii) For each StoNet estimate $\hat{\boldsymbol{\theta}}^{(t)} \in \mathcal{S}$, construct the prediction interval

$$\left(\mu_j(\boldsymbol{z}, \hat{\boldsymbol{\theta}}^{(t)}) - 1.96\sqrt{\widehat{\Sigma}_{h+1,j}^{(t)} + \hat{\varsigma}_{h+1,j}^{2(t)}}, \mu_j(\boldsymbol{z}, \hat{\boldsymbol{\theta}}^{(t)}) + 1.96\sqrt{\widehat{\Sigma}_{h+1,j}^{(t)} + \hat{\varsigma}_{h+1,j}^{2(t)}}\right), \quad (7)$$

where $\widehat{\Sigma}_{h+1,j}^{(t)}$ denotes the $(j,j)$-th diagonal element of $\widehat{\Sigma}_{h+1}^{(t)}$.

(iii) Output the final 95% prediction interval of $\mu_j(\boldsymbol{z}, \boldsymbol{\theta}^*)$ by averaging $m$ intervals obtained in step (ii).

The above procedure can be easily extended to the StoNet with a logistic regression output layer via the Wald/endpoint transformation. By Lemma A2, the above procedure can also be applied to the StoNet trained by the adaptive stochastic gradient MCMC algorithm.

## 5    An Illustrative Example

This example serves as a validation of our theoretical results. Consider two models:

$$y = 2\psi(2x_1 - x_2) + 2\psi(x_3 - 2x_4) - \psi(2x_5) + 0x_6 + \cdots + 0x_{20} + \epsilon, \quad (8)$$
$$y = \psi(2\psi(2x_1 - x_2)) + 2\psi(2\psi(x_3 - 2x_4) - \psi(2x_5)) + 0x_6 + \cdots + 0x_{20} + \epsilon, \quad (9)$$

where $\psi(\cdot) = \tanh(\cdot)$, $\epsilon \sim N(0,1)$, $\boldsymbol{x} = (x_1, x_2, \ldots, x_{20})$, $x_i \sim N(0,1)$ for $i = 1, 2, \ldots, 20$, and $x_i$'s are correlated with a mutual correlation coefficient of 0.5. Equations (8) and (9) represent neural networks with one and two hidden layers, respectively. For both models, the variables $x_1, x_2, \ldots, x_5$ are true and the others are false. The strong mutual correlation makes the true variables hard to identify. From each model, we simulated 1000 samples, 500 samples for training and 500 samples for test. We use this example to examine the performance of the StoNet in nonlinear variable selection and prediction uncertainty quantification, and to examine the effect of $\boldsymbol{\sigma}^2 = (\sigma_1^2, \ldots, \sigma_{h+1}^2)$ on the performance of the StoNet as well. To make the tests more convincing, we particularly generated the data from true DNN models.

We modeled the data from model (8) by a StoNet with structure 20-500-1, and that from model (9) by a StoNet with structure 20-500-100-1. We trained the StoNets by Algorithm 2 with different parameter settings as given in the appendix. The major differences of the settings are at the values of $\sigma_i$'s and, for convenience, we call the settings by half-$\sigma^2$, single-$\sigma^2$ and double-$\sigma^2$, respectively. Under each setting, Algorithm 2 was run for 2000 epochs with the Lasso penalty $P_\lambda(\boldsymbol{\theta}) = \lambda\|\boldsymbol{\theta}\|_1$. Various values of $\lambda$ have been tried as described below.

To examine the effect of $\lambda$, we mimic the regularization path for LASSO and measure the importance of each variable by the average output gradient $\frac{1}{n}\sum_{k=1}^n \frac{\partial \hat{\mu}(\boldsymbol{x})}{\partial x_i}|_{\boldsymbol{x}^{(k)}}$ calculated over training samples, where $\hat{\mu}(\boldsymbol{x})$ denotes the forward prediction function of the StoNet and $\boldsymbol{x}^{(k)}$ denotes the $k$th-observation of the training set. Using the partial derivative to evaluate the dependence of a function on a particular variable has been proposed by RosascoLorenzo et al. (2013), and employed in Zheng et al. (2020) for sparse graphical modeling. Figure 2 (a) shows the regularization path of the StoNet for the model (9). The path for the model (8) is shown in Figure A1(a). Further, we examined the paths of each variable and found that for both models (8) and (9), the true variables $x_1, x_2, \ldots, x_5$ can be correctly identified by the StoNet with appropriate values of $\lambda$.

(a) 2-hidden-layer StoNet                  (b) 2-hidden-layer DNN

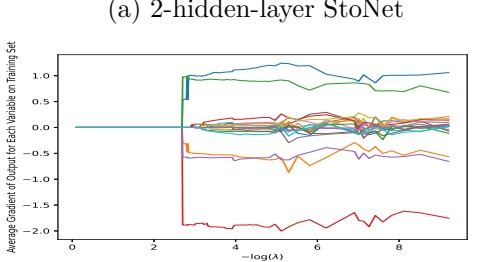 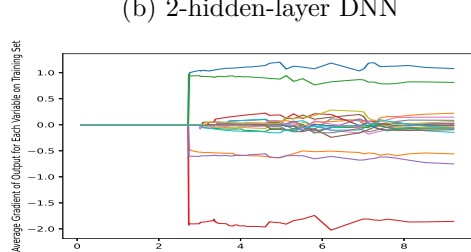

Figure 2: Variable selection paths by the StoNet (under the single-$\sigma^2$ setting) and DNN for the model (9), where $y$-axis is the average output gradient $\frac{1}{n}\sum_{k=1}^n \frac{\partial \hat{\mu}(\boldsymbol{x})}{\partial x_i}|_{\boldsymbol{x}^{(k)}}$ calculated over the training data, and $x$-axis is $-\log(\lambda)$. The 5 lines with $\frac{1}{n}\sum_{k=1}^n \frac{\partial \hat{\mu}(\boldsymbol{x})}{\partial x_i}|_{\boldsymbol{x}^{(k)}}$ away from 0 corresponds to the 5 true variables

Corollary 1 shows that consistent sparse deep learning can also be achieved by training the DNN with a Lasso penalty. To illustrate this result, we considered two DNNs with the same structures as the StoNets used above, imposed the Lasso penalty on their weights, and trained them using SGD with momentum. We trained each DNN for 2000 epochs with a constant learning rate of 0.001, a momentum parameter of 0.9 and a mini-batch size of 50. Figure 2(b) show the variable selection path of the DNN for the model (9). The path for the model (8) can be found in Figure A1. As expected, the true variables can also be identified by the sparse DNN trained with the Lasso penalty.

Table 1: Coverage rates of 95% prediction intervals produced by the StoNet for 500 test samples simulated from the models (8) and (9), where the number in the parentheses represents the standard deviation of the coverage rate.

| Model | half-$\sigma^2$ | single-$\sigma^2$ | double-$\sigma^2$ |
|---|---|---|---|
| Model (8) | 94.766% (2.157%) | 94.496% (2.162%) | 94.310% (2.197%) |
| Model (9) | 94.642% (2.189%) | 94.396% (2.256%) | 94.300% (2.290%) |

Next, we examined the performance of the StoNet in prediction uncertainty quantification. We generated 100 training datasets, each consisting of 500 samples, from each of the models (8) and (9). For each training dataset, a StoNet was trained as described above, and a prediction interval was constructed for each sample point of the test dataset with the StoNet estimate obtained at the last iteration of the run. Table 1 summarizes the coverage rates of the predictive intervals by averaging the coverage status of $500 \times 100$ prediction intervals, where '500' refers to the total number of test points and '100' refers to the total number of training datasets. As expected, the StoNet produces better coverage rates with smaller values

of $\boldsymbol{\sigma}^2$, since the true models are DNN models. Figure A2 shows the prediction intervals produced by the StoNet at some test points.

## 6  Statistical Inference for Deep Learning

This section discusses two applications of StoNet in statistical inference for deep learning models. One is for identifying important features for the response or ranking their importance. The other is for quantifying prediction uncertainty through a post-StoNet modeling procedure.

### 6.1  Identification of Important Input Features

As a further illustration for this type of applications, we consider a big dataset, *CoverType*, available at UCI repository. It consists of $n = 581,012$ samples and $p = 54$ features, collected for classification of forest cover types from cartographic variables. We used 80% of the data for training and the other 20% for testing. We fit the data by a DNN of two hidden layers, with 1000 and 500 hidden units, respectively. Refer to Section G for detailed settings of training parameters. Figure 3 indicates important input features can be identified for the DNN along the path of parameter regularization as in applications of Lasso for linear models.

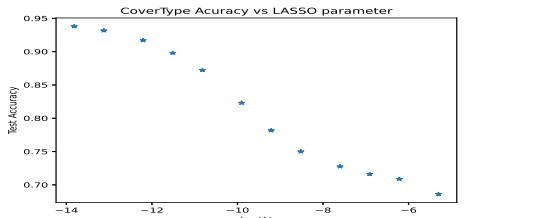 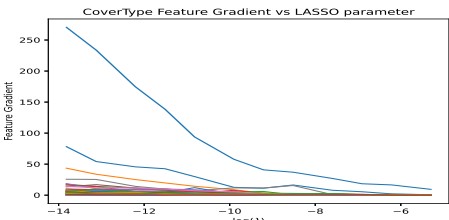

Figure 3: Test accuracy (left) and feature gradient (right) versus regularization parameters for the CoverType data.

### 6.2  Post-StoNet Modeling for Uncertainty Quantification

In real applications, use of large-scale deep neural networks, such as residual networks (He et al., 2016) and transformer (Dosovitskiy et al., 2020), has been a common practice. However, these large-scale models can be miscalibrated (Guo et al., 2017). To address this issue, we propose a post-StoNet modeling procedure, without significantly changing the current practice of large-scale models. The proposed procedure is as follows:

  (i) Transform the explanatory variables by calculating the output of the last-hidden-layer of a well-trained DNN.

  (ii) Learn a simple sparse StoNet(e.g. with one hidden layer only) using the transformed data and their response on the validation data.

The uncertainty quantification follows the procedure given in Section 4, except that the $z$ variable is given by the transformed data. We provide an intuitive justification for this procedure: as shown in Liang et al. (2022), the last-hidden-layer's output of the StoNet serves as a nonlinear sufficient dimension reduction (SDR) of the input data. Building upon the asymptotic equivalence between the StoNet and DNN (Lemma 1), the transformed data from a well-trained DNN approximates a sufficient dimension reduction of the input data. The DNN model typically gives simple linear relationship between transformed data and response, but the linear relationship may not hold anymore on the validation set due to the possible over-fitting issue. Therefore, we remodel it using a simple sparse StoNet, which enables the prediction uncertainty to be correctly quantified. In what follows, we use numerical experiments to show that the proposed procedure improves model calibration and provide shorter confidence intervals compared to the conformal method.

**Classification Problems**   We conduct experiments on CIFAR10 data. Following the setting of post-calibration methods in Guo et al. (2017), we split the training data into a training set of 45000 images and a hold out validation set of 5000 images. We modeled the data using DenseNet40(Huang et al., 2017), ResNet110(He et al., 2016) and WideResNet-28-10(Zagoruyko & Komodakis, 2016). Refer to Section G for detailed settings of the training

parameters. For comparison, we also applied temperature scaling and matrix scaling (Guo et al., 2017) to the same trained models. We repeated the experiment 10 times and report the mean and standard deviation of accuracy (ACC), negative log-likelihood loss (NLL) and expected calibration error (ECE) in Table 2. The result shows that the post-StoNet modeling method significantly improves model calibration, especially in terms of ECE.

Table 2: Calibration results for CIFAR10 data, where standard deviations of the respective measures are given in parentheses.

| Network | Size | Method | ACC | NLL | ECE |
|---|---|---|---|---|---|
| DenseNet40 | 176K | No Post Calibration | **92.88%(0.19%)** | 0.3076(0.0094) | 0.0434(0.0019) |
| | | Matrix Scaling | 92.73%(0.20%) | 0.2226(0.0052) | 0.0132(0.0026) |
| | | Temp. Scaling | **92.88%**(0.19%) | 0.2194(0.0055) | 0.0117(0.0016) |
| | | Post-StoNet | 92.79%(0.17%) | **0.2175**(0.0034) | **0.0047**(0.0010) |
| ResNet110 | 1.7M | No Post Calibration | **93.23%(0.37%)** | 0.3113(0.0220) | 0.0444(0.0030) |
| | | Matrix Scaling | 92.96%(0.32%) | 0.2127(0.0097) | 0.0145(0.0023) |
| | | Temp. Scaling | **93.23%**(0.37%) | 0.2077(0.0092) | 0.0122(0.0016) |
| | | Post-StoNet | 93.22%(0.31%) | **0.2045**(0.0086) | **0.0070**(0.0014) |
| WideResNet-28-10 | 36M | No Post Calibration | **95.76%(0.13%)** | 0.1710(0.0077) | 0.0258(0.0014) |
| | | Matrix Scaling | 95.71%(0.14%) | 0.1475(0.0042) | 0.0104(0.0014) |
| | | Temp. Scaling | **95.76%**(0.13%) | 0.1489(0.0055) | 0.0120(0.0017) |
| | | Post-StoNet | 95.63%(0.08%) | **0.1448**(0.0031) | **0.0089**(0.0007) |

**Regression Problems** We used 4 datasets from UCI repository ranging in size from thousands to hundreds of thousands. For each dataset, we first train a DNN model on training set then applied the post-StoNet procedure to generate 90% prediction intervals(see appendix G.4 for details). For comparison, we applied the split conformal method (Vovk et al., 2005) to the same trained DNNs. The results in Table 3 demonstrate a significant improvement in terms of the lengths of the prediction confidence intervals This improvement is largely attributed to the well-trained DNNs, which transform the potentially highly nonlinear mapping (from inputs to response) into a relatively simple mapping (from last-hidden-layer outputs to response). The sparsity of the post-StoNet mitigates potential overfitting issues suffered by the DNNs, thus enhancing prediction performance. However, adapting with overfitting has been beyond the ability of the conformal method.

Table 3: Average coverage rate and confidence interval length of test sets of 20 random split of data. The standard deviations are given in the parentheses.

| Dataset | N | P | Model | Coverage Rate | Interval length |
|---|---|---|---|---|---|
| Wine | 1,599 | 11 | Post-StoNet | 0.9042(0.0126) | **2.0553**(0.0719) |
| | | | Split Conformal | 0.8958(0.0302) | 2.4534(0.1409) |
| Power Plant | 9,568 | 4 | Post-StoNet | 0.9109(0.0070) | **13.4726**(0.2420) |
| | | | Split Conformal | 0.8999(0.0082) | 14.5719(0.2676) |
| Protein | 45,730 | 9 | Post-StoNet | 0.8941(0.0028) | **13.1319**(0.0494) |
| | | | Split Conformal | 0.9004(0.0022) | 14.4296(0.0886) |
| Year | 515,345 | 90 | Post-StoNet | 0.9064(0.0013) | **29.4272**(0.0923) |
| | | | Split Conformal | 0.9001(0.0010) | 32.1068(0.3726) |

## 7 CONCLUSION

We have demonstrated that the StoNet effectively bridges the gap between linear models and deep learning, allowing us to adapt theories and methods developed for linear models to deep learning models. Specifically, we have adapted sparse learning theory from linear models to DNNs, enabling the identification of important input features in DNN training with the Lasso penalty. We have also employed the StoNet to quantify uncertainty in DNN predictions. Our numerical results suggest that the StoNet significantly improves prediction uncertainty quantification for deep learning models compared to the conformal method and other post processing calibration methods.

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

# Appendix

## A Assumptions for Lemma 1

The property of the StoNet as an approximator to the DNN, i.e., asymptotically they have the same loss function as the training sample size $n \to \infty$, has been studied in Liang et al. (2022). A brief review for their theory is provided as follows, which form the basis for this work.

Let $\boldsymbol{\theta} = (\boldsymbol{w}_1, \boldsymbol{b}_1, \ldots, \boldsymbol{w}_{h+1}, \boldsymbol{b}_{h+1})$ denote the collection of all weights of the StoNet (2), let $\boldsymbol{\Theta}$ denote the space of $\boldsymbol{\theta}$, let $\boldsymbol{Y}_{\mathrm{mis}} = (\boldsymbol{Y}_1, \boldsymbol{Y}_2, \ldots, \boldsymbol{Y}_h)$ denote the collection of all latent variables, let $\pi(\boldsymbol{Y}, \boldsymbol{Y}_{\mathrm{mis}}|\boldsymbol{X}, \boldsymbol{\theta})$ denote the likelihood function of the StoNet, and let $\pi_{\mathrm{DNN}}(\boldsymbol{Y}|\boldsymbol{X}, \boldsymbol{\theta})$ denote the likelihood function of the DNN model (1). Regarding the network structure, activation function and the variance of the latent variables, they made the following assumption:

**Assumption A1.** *(i) $\boldsymbol{\Theta}$ is compact, i.e., $\boldsymbol{\Theta}$ is contained in a $d_\theta$-ball centered at 0 with radius $r$; (ii) $\mathbb{E}(\log \pi(\boldsymbol{Y}, \boldsymbol{Y}_{\mathrm{mis}}|\boldsymbol{X}, \boldsymbol{\theta}))^2 < \infty$ for any $\boldsymbol{\theta} \in \boldsymbol{\Theta}$; (iii) the activation function $\psi(\cdot)$ is $c'$-Lipschitz continuous for some constant $c'$; (iv) the network's depth $h$ and widths $d_l$'s are both allowed to increase with $n$; (v) $\sigma_1 \leq \sigma_2 \leq \cdots \leq \sigma_{h+1}$, $\sigma_{h+1} = O(1)$, and $d_{h+1}(\prod_{i=k+1}^{h} d_i^2)d_k\sigma_k^2 \prec \frac{1}{h}$ for any $k \in \{1, 2, \ldots, h\}$.*

Assumption A1-(iii) allows the StoNet to work with a wide range of Lipschitz continuous activation functions such as *tanh*, *sigmoid* and *ReLU*. Assumption A1-(v) constrains the size of noise added to each hidden neuron, where the factor $d_{h+1}(\prod_{i=k+1}^{h} d_i^2)d_k$ can be understood as the amplification factor of the noise $\boldsymbol{e}_k$ at the output layer. In general, the noise added to the first few hidden layers should be small to prevent large random errors propagated to the output layer. Under Assumption A1, they proved part (i) of Lemma 1.

Further, regarding the equivalence between training the StoNet and the DNN, they made the following assumption regarding the energy surface of the DNN. Let $Q^*(\boldsymbol{\theta}) = \mathbb{E}(\log \pi_{\mathrm{DNN}}(\boldsymbol{Y}|\boldsymbol{X}, \boldsymbol{\theta}))$, where the expectation is taken with respect to the joint distribution $\pi(\boldsymbol{X}, \boldsymbol{Y})$. By Assumption A1-(i)&(ii) and the law of large numbers,

$$\frac{1}{n}\sum_{i=1}^{n} \log \pi_{\mathrm{DNN}}(\boldsymbol{Y}^{(i)}|\boldsymbol{X}^{(i)}, \boldsymbol{\theta}) - Q^*(\boldsymbol{\theta}) \xrightarrow{p} 0 \tag{A1}$$

holds uniformly over $\Theta$. They assumed $Q^*(\boldsymbol{\theta})$ satisfies the following regularity conditions:

**Assumption A2.** *(i)$Q^*(\boldsymbol{\theta})$ is continuous in $\boldsymbol{\theta}$ and uniquely maximized at $\boldsymbol{\theta}^*$; (ii) for any $\epsilon > 0$, $\sup_{\boldsymbol{\theta} \in \Theta \backslash B(\epsilon)}Q^*(\boldsymbol{\theta})$ exists, where $B(\epsilon) = \{\boldsymbol{\theta} : \|\boldsymbol{\theta} - \boldsymbol{\theta}^*\| < \epsilon\}$, and $\delta = Q^*(\boldsymbol{\theta}^*) - \sup_{\boldsymbol{\theta} \in \Theta \backslash B(\epsilon)}Q^*(\boldsymbol{\theta}) > 0$.*

Assumption A2 restricts the shape of $Q^*(\boldsymbol{\theta})$ around the global maximizer, which cannot be discontinuous or too flat. Given nonidentifiability of the neural network model, Assumption A2 has implicitly assumed that each $\boldsymbol{\theta}$ is unique up to the loss-invariant transformations, e.g., reordering the hidden neurons of the same hidden layer and simultaneously changing the signs of some weights and biases. Under Assumptions A1 and A2, they proved part (ii) of Lemma 1.

## B The IRO Algorithm for training StoNet

Algorithm 1 gives the IRO algorithm(Liang et al., 2018) for training StoNet.

## C Proofs of Theorem 1

In order to establish the consistency of the sparse StoNet, we need the following assumptions. Following Meinshausen & Yu (2009), we define the *m*-sparse minimal and maximal eigenvalues

---

**Algorithm 1** IRO Algorithm for StoNet; Liang et al. (2018)

---

**Input**: Dataset $(\boldsymbol{X}, \boldsymbol{Y})$, total iteration number $T$, and Monte Carlo step number $t_{MC}$.

**Initialization**: Randomly initialize the network parameters $\hat{\boldsymbol{\theta}}^{(0)} = (\hat{\boldsymbol{\theta}}_1^{(0)}, \ldots, \hat{\boldsymbol{\theta}}_{h+1}^{(0)})$.

**for** $t = 1$ **to** $T$ **do**

    • **Imputation step**: For each sample $(\boldsymbol{X}^{(i)}, \boldsymbol{Y}^{(i)})$, draw $\boldsymbol{Y}_{\text{mis}}^{(i,t)}$ from $\pi(\boldsymbol{Y}_{\text{mis}}|\boldsymbol{Y}^{(i)}, \boldsymbol{X}^{(i)}, \hat{\boldsymbol{\theta}}_n^{(t-1)}, \boldsymbol{\sigma}_n^2)$ with a Metropolis or Langevin dynamics kernel by iterating for $t_{MC}$ steps.

    • **Regularized optimization step**: Based on the pseudo-complete data $(\boldsymbol{Y}, \boldsymbol{Y}_{\text{mis}}^{(t)}, \boldsymbol{X})$, update $\hat{\boldsymbol{\theta}}_n^{(t-1)}$ by minimizing a penalized loss function, i.e., setting

$$\hat{\boldsymbol{\theta}}_n^{(t)} = \arg \min_{\boldsymbol{\theta}} \left\{ -\frac{1}{n} \sum_{i=1}^{n} \log \pi(\boldsymbol{Y}^{(i)}, \boldsymbol{Y}_{\text{mis}}^{(i,t)} | \boldsymbol{X}^{(i)}, \boldsymbol{\theta}, \boldsymbol{\sigma}_n^2) + P_{\lambda_n}(\boldsymbol{\theta}) \right\}, \tag{A2}$$

where the penalty function $P_{\lambda_n}(\boldsymbol{\theta})$ is chosen such that $\hat{\boldsymbol{\theta}}_n^{(t)}$ forms a consistent estimator of

$$\boldsymbol{\theta}_*^{(t)} = \arg \max_{\boldsymbol{\theta}} \mathbb{E}_{\boldsymbol{\theta}_n^{(t-1)}} \log \pi(\boldsymbol{Y}, \boldsymbol{Y}_{\text{mis}} | \boldsymbol{X}, \boldsymbol{\theta}, \boldsymbol{\sigma}_n^2)$$

$$= \arg \max_{\boldsymbol{\theta}} \int \log \pi(\boldsymbol{Y}_{\text{mis}}, \boldsymbol{Y} | \boldsymbol{X}, \boldsymbol{\theta}, \boldsymbol{\sigma}_n^2) \pi(\boldsymbol{Y}_{\text{mis}} | \boldsymbol{Y}, \boldsymbol{X}, \boldsymbol{\theta}_n^{(t-1)}, \boldsymbol{\sigma}_n^2) \pi(\boldsymbol{Y} | \boldsymbol{X}, \boldsymbol{\theta}^*, \boldsymbol{\sigma}_n^2) d\boldsymbol{Y}_{\text{mis}} d\boldsymbol{Y}, \tag{A3}$$

where $\boldsymbol{\theta}_*^{(t)}$ is called the working true parameter at iteration $t$.

**end for**

**Output**: $\hat{\theta}_n^{(T)}$.

---

for a matrix $\Sigma$ as follows:

$$\phi_{\min}(m|\Sigma) = \min_{\boldsymbol{\beta}: \|\boldsymbol{\beta}\|_0 \leq m} \frac{\boldsymbol{\beta}^T \Sigma \boldsymbol{\beta}}{\boldsymbol{\beta}^T \boldsymbol{\beta}},$$

$$\phi_{\max}(m|\Sigma) = \max_{\boldsymbol{\beta}: \|\boldsymbol{\beta}\|_0 \leq m} \frac{\boldsymbol{\beta}^T \Sigma \boldsymbol{\beta}}{\boldsymbol{\beta}^T \boldsymbol{\beta}},$$

which represent, respectively, the minimal and maximal eigenvalues of any $m \times m$-dimensional principal submatrix. Let $\boldsymbol{\Sigma}_n \in \mathbb{R}^{p_n \times p_n}$ denote the covariance matrix of the input variables. Let $q_{l,k,n}^{(t)}$ denotes the size of the working true regression formed for neuron $k$ of layer $l$ at iteration $t$, as implied by the working true parameter $\boldsymbol{\theta}_*^{(t)}$.

**Assumption A3.** *(i) The input variable $\boldsymbol{X}$ is bounded, and there exist a constants $0 < \kappa_{\min} < \infty$ such that $\liminf_{n \to \infty} \phi_{\min}(\min\{n, p_n\}|\boldsymbol{\Sigma}_n) \geq \kappa_{\min}$; (ii) there exists a sparse exponent $s \in [0,1]$ such that $q_{l,k,n}^{(t)} \prec d_{l-1,n}^s$ for $1 \leq k \leq d_{l,n}$ $1 \leq l \leq h_n + 1$ and any iteration $t$, and set $(\sigma_{1,n}^2, \sigma_{2,n}^2, \ldots, \sigma_{h_n+1,n}^2)$ such that the following conditions hold: $\frac{\kappa_{\min}^2}{\sigma_{1,n}^2} \succ \frac{h_n d_{1,n} p_n^s \log p_n}{n}$, and $\frac{\sigma_{l-1,n}^4}{\sigma_{l,n}^2} \succ \frac{h_n d_{l,n} d_{l-1,n}^s \log d_{l-1,n}}{n}$ for any $l \in \{2, 3, \ldots, h_n + 1\}$; (iii) the activation function $\Psi(\cdot)$ is bounded.*

Assumption A3-(i) is regular, which has often been used in the literature of high-dimensional variable selection, see e.g., Huang et al. (2008). Assumption A3-(ii) works with Assumption A1-(v) to constrain the range of $\sigma_{l,n}$'s. We note that such a uniform sparse exponent $s$ always exists, which can be equal to 1 in the worst scenario. Assumption A3-(iii) is more or less a technical condition. Since $\sigma_{l,n}^2$'s are usually set to very small values, it is easy to restrict the random errors $\boldsymbol{e}_i$'s to a compact space with high probability. Therefore, an unbounded activation function such as *ReLU* can still be used in the StoNet, but the following theoretical results need to be slightly modified to hold with high probability.

Regarding the setting of regularization parameters, we have the following assumption which directly follows from the theory developed by Meinshausen & Yu (2009) for linear regression and Huang et al. (2008) for logistic regression.

**Assumption A4.** *The Lasso penalty is used for the StoNet. At each iteration t, (i) set the regularization parameter $\lambda_{l,n}^{(t)} \asymp \sigma_{l,n}(n \log d_{l-1,n})^{1/2}$ for each linear regression layer l; and (ii) set the regularization parameter $\lambda_{l,n}^{(t)} \asymp (n^{2+\varepsilon} \log d_{l-1,n})^{1/3}$ for some $\varepsilon \in (0,1)$ for each logistic regression layer.*

In order to prove Theorem 1, we first introduce the following lemma

**Lemma A1.** *For any $L \in \{1, 2, \ldots, h\}$, let $\Sigma_L^{(t)}$ denote the sample covariance matrix of the covariates of the linear regressions formed for each neuron of layer $L+1$ at iteration t. If Assumption A1 and Assumption A3 hold, then there exist constants $c > 0$ and $0 < \kappa_{\max,L} < \infty$ such that for any iteration t,*

$$\phi_{\min}(\min\{n, d_{L,n}\}|\Sigma_L^{(t)}) \geq c\sigma_{L,n}^2, \quad \phi_{\max}(\min\{n, d_{L,n}\}|\Sigma_L^{(t)}) \leq \kappa_{\max,L}.$$

*Proof.* For simplicity of notations, we suppress the iteration index $t$. Let $\tilde{Y}_L = b_L + w_L \Psi(Y_{L-1})$ for $L = 2, \ldots, h$, and let $\tilde{Y}_1 = b_1 + w_1 X$. By the definition of the StoNet model (2), $Y_L$ can be written as $Y_L = \tilde{Y}_L + e_L$ for $L \in \{1, 2, \ldots, h\}$.

Since $\sigma_L^2$ has been set to a very small value, we have $\Psi(Y_L) \approx \Psi(\tilde{Y}_L) + \nabla_{\tilde{Y}_L} \Psi(\tilde{Y}_L) \circ e_L$, where $\circ$ denotes elementwise product. Then

$$\begin{aligned}
\Sigma_L &\approx \mathrm{Var}(\mathbb{E}(\Psi(\tilde{Y}_L) + \nabla_{\tilde{Y}_L} \Psi(\tilde{Y}_L) \circ e_L | \tilde{Y}_L)) + \mathbb{E}(\mathrm{Var}(\Psi(\tilde{Y}_L) + \nabla_{\tilde{Y}_L} \Psi(\tilde{Y}_L) \circ e_L | \tilde{Y}_L)) \\
&= \mathrm{Var}(\Psi(\tilde{Y}_L)) + \mathrm{diag}\left\{\sigma_L^2 \mathbb{E}[\nabla_{\tilde{Y}_L} \Psi(\tilde{Y}_L) \circ \nabla_{\tilde{Y}_L} \Psi(\tilde{Y}_L)]\right\},
\end{aligned} \tag{A4}$$

where $\mathrm{diag}\{v\}$ with $v \in \mathbb{R}^d$ denotes a $d \times d$ diagonal matrix with diagonal elements being $v$.

By Assumption A3-(iii), the activation function is bounded. For example, *tanh* or *sigmoid* is used in the model. By Assumption A1, there exists some constant $C_1$ such that $\|b_L\|_\infty < C_1, \|w_L\|_\infty < C_1$. By Assumption A3, $\|X\|_\infty$ is bounded. Therefore, there exists some constant $C_2$ such that for any $L \in \{1, 2, \ldots, h\}$, $\|\tilde{Y}_L\|_\infty \leq C_1 + C_1 C_2$ holds by rescaling $X$ by a factor of $\prod_{l=1}^h d_l$. Since both $\Psi(\tilde{Y}_L)$ and $\nabla_{\tilde{Y}_L} \Psi(\tilde{Y}_L)$ are bounded, there exists a constant $\kappa_{\max,L}$ such that

$$\phi_{\max}(d_{L,n}|\Sigma_L) \leq \kappa_{\max,L}.$$

To establish the lower bound, we note that $\|\tilde{Y}_L\|_\infty \leq C_1 + C_1 C_2$. Therefore, for an activation function which has nonzero gradients on any closed interval, e.g., *tanh* and *sigmoid*, there exists a constant $C_3 > 0$ such that $\min_{i=1,\ldots,d_L} \nabla_{\tilde{Y}_L} \Psi(\tilde{Y}_L)_i > C_3$, where $\nabla_{\tilde{Y}_L} \Psi(\tilde{Y}_L)_i$ denotes the $i$-th element of $\nabla_{\tilde{Y}_L} \Psi(\tilde{Y}_L)$. Then we can take $\kappa_{\min,L} = \sigma_L^2 C_3^2$ such that

$$\phi_{\min}(d_{L,n}|\Sigma_L) \geq \kappa_{\min,L},$$

which completes the proof.

$\square$

**Proof of Part (i) of Theorem 1**

*Proof.* By Lemma A1, $\Sigma_L^{(t)}$ satisfies the requirements of Theorem 1 of Meinshausen & Yu (2009) and Theorem 1 of Huang et al. (2008). Then, by Theorem 1 of Meinshausen & Yu (2009) (for linear regression) and Theorem 1 of Huang et al. (2008) (for logistic regression), we have $r_n$ as given in the lemma by summarizing the $l_2$-errors of coefficient estimation for all $\sum_{l=1}^{h+1} d_l$ regression/logistic regressions. Further, by the setting of $(\sigma_{1,n}^2, \ldots, \sigma_{h+1,n}^2)$ as specified in Assumption A3, we have $r_n \to 0$ as $n \to \infty$. This completes the proof of part (i) of Theorem 1. $\square$

Further, let's consider the mapping $M(\theta)$ as defined in (A3), i.e., $M(\theta) = \arg\max_{\theta'} \mathbb{E}_\theta \log \pi(Y, Y_{\mathrm{mis}}|X, \theta', \sigma_n^2)$. As argued in Liang et al. (2018) and Nielsen (2000), it is reasonable to assume that the mapping is a contraction, as a recursive application of the mapping, i.e., setting $\theta_n^{(t+1)} = \theta_*^{(t+1)} = M(\theta_n^{(t)})$, leads to a monotone increase of the target expectations $\mathbb{E}_{\theta_n^{(t)}} \log \pi(Y, Y_{\mathrm{mis}}|X, \theta_n^{(t+1)}, \sigma_n^2)$ for $t = 1, 2, \ldots$.

**Assumption A5.** *The mapping $M(\boldsymbol{\theta})$ is differentiable. Let $\lambda_n(\boldsymbol{\theta})$ be the largest singular value of $\partial M(\boldsymbol{\theta})/\partial \boldsymbol{\theta}$. There exists a number $\lambda^* < 1$ such that $\lambda_n(\boldsymbol{\theta}) \le \lambda^*$ for all $\boldsymbol{\theta} \in \Theta_n$ for sufficiently large $n$ and almost every $D_n$ observation sequence.*

**Proof of Part (ii) of Theorem 1**

*Proof.* Then part (ii) of Theorem 1 directly follows from Theorem 4 of Liang et al. (2018) that the estimator $\hat{\boldsymbol{\theta}}_n^{(t)}$ is consistent when both $n$ and $t$ are sufficiently large. □

To establish the structure selection consistency in Part (iii) of Theorem 1, we need the following $\theta$-min condition:

**Assumption A6.** *($\theta$-min condition) $\min_{k \in \boldsymbol{\gamma}^*} |\theta_k^*| \succ \sqrt{r_n}$, where $\boldsymbol{\gamma}^* = \{k : \theta_k^* \ne 0\}$ is the set of indexes of non-zero elements of $\boldsymbol{\theta}^*$ and $\theta_k^*$ denotes the $k$-th component of $\boldsymbol{\theta}^*$.*

Assumption A6 is essentially an identifiability condition, which ensures the non-zero elements of $\boldsymbol{\theta}^*$ can be distinguished from 0. This is a typical assumption for high-dimensional variable selection, see e.g., Zhao & Yu (2006). Under Assumption A6, the proof of Part (iii) of Theorem 1 is given as follows:

**Proof of Part (iii) of Theorem 1**

*Proof.* Let $\hat{\boldsymbol{\theta}}_n^{(t)}$ denote the estimate of $\boldsymbol{\theta}_n$ at iteration $t$, and let $\boldsymbol{\theta}_*^{(t)}$ denote its "true" value at iteration $t$, and let $\boldsymbol{\theta}^*$ denote its true value in the StoNet. By the proof of Theorem 4 of Liang et al. (2018) and Theorem 1 of Meinshausen & Yu (2009), for the StoNet with the linear regression output layer, we have

$$\mathbb{E}\|\hat{\boldsymbol{\theta}}_n^{(t)} - \boldsymbol{\theta}^*\| \le \frac{1}{1-\lambda^*}\mathbb{E}\|\hat{\boldsymbol{\theta}}_n^{(t)} - \boldsymbol{\theta}_*^{(t)}\| \prec \frac{\sqrt{r_n}}{1-\lambda^*}, \quad \text{as } t \to \infty, \tag{A5}$$

by summarizing all $d_1 + d_2 + \cdots + d_{h+1}$ linear regressions, where $\lambda^*$ is a constant as defined in Assumption A5. For the StoNet with the logistic regression output layer, we have the same result by Theorem 1 of Huang et al. (2008). Further, by Markov inequality, there exists a constant $c$ such that

$$P\left(\|\hat{\boldsymbol{\theta}}_n^{(t)} - \boldsymbol{\theta}^*\| > c\sqrt{r_n}\right) \to 0, \quad \text{as } n \to \infty \text{ and } t \to \infty.$$

Then, by Assumption A6,

- For any $i \in \boldsymbol{\gamma}^*$, $\|\hat{\boldsymbol{\theta}}_n^{(t)} - \boldsymbol{\theta}^*\| \le c\sqrt{r_n}$ implies $|\hat{\boldsymbol{\theta}}_{i,n}^{(t)}| > c\sqrt{r_n}$.

- For any $i \notin \boldsymbol{\gamma}^*$, $\|\hat{\boldsymbol{\theta}}_n^{(t)} - \boldsymbol{\theta}^*\| \le c\sqrt{r_n}$ implies $|\hat{\boldsymbol{\theta}}_{i,n}^{(t)}| < c\sqrt{r_n}$.

Therefore,

$$P(\hat{\boldsymbol{\gamma}} = \boldsymbol{\gamma}^*) \ge P((\|\hat{\boldsymbol{\theta}}_n^{(t)} - \boldsymbol{\theta}^*\| \le c\sqrt{r_n}) \to 1, \quad \text{as } n \to \infty \text{ and } t \to \infty, \tag{A6}$$

which concludes the proof. □

## D   Adaptive Stochastic Gradient MCMC for Efficient StoNet Learning

### D.1   Adaptive stochastic gradient Hamilton Monte Carlo

The IRO algorithm is developed under the full data setting and thus less scalable with respect to big data. To address this issue, we suggest to train the sparse StoNet using an adaptive stochastic gradient MCMC(ASGMCMC) algorithm by Liang et al. (2022), which is scalable with respect to big data by making use of mini-batch samples at each iteration. To make the paper self-contained, we gives a review of ASGMCMC algorithm below.

Let $\pi(\boldsymbol{Y}|\boldsymbol{X},\boldsymbol{\theta},\boldsymbol{\sigma}^2) = \int \pi(\boldsymbol{Y},\boldsymbol{Y}_{\mathrm{mis}}|\boldsymbol{X},\boldsymbol{\theta},\boldsymbol{\sigma}^2)d\boldsymbol{Y}_{\mathrm{mis}}$ denote the likelihood function of the observed data for the StoNet. By Fisher's identity, we have

$$\nabla_{\boldsymbol{\theta}} \log \pi(\boldsymbol{Y}|\boldsymbol{X},\boldsymbol{\theta},\boldsymbol{\sigma}^2) = \int \nabla_{\boldsymbol{\theta}} \log \pi(\boldsymbol{Y},\boldsymbol{Y}_{\mathrm{mis}}|\boldsymbol{X},\boldsymbol{\theta},\boldsymbol{\sigma}^2)\pi(\boldsymbol{Y}_{\mathrm{mis}}|\boldsymbol{Y},\boldsymbol{X},\boldsymbol{\theta},\boldsymbol{\sigma}^2)d\boldsymbol{Y}_{\mathrm{mis}}.$$

Therefore, the sparse StoNet can also be trained by solving the equation

$$\int \nabla_{\boldsymbol{\theta}}[\log \pi(\boldsymbol{Y},\boldsymbol{Y}_{\mathrm{mis}}|\boldsymbol{X},\boldsymbol{\theta},\boldsymbol{\sigma}^2) + \log P_{\lambda}(\boldsymbol{\theta})]\pi(\boldsymbol{Y}_{\mathrm{mis}}|\boldsymbol{Y},\boldsymbol{X},\boldsymbol{\theta},\boldsymbol{\sigma}^2)d\boldsymbol{Y}_{\mathrm{mis}} = 0, \qquad \text{(A7)}$$

where $P_{\lambda}(\boldsymbol{\theta})$ denotes a penalty function satisfying Assumption A4. By Theorem 1 of Liang et al. (2018), solving for (A7) will lead to the same solution as solving the optimization problem specified in (5).

By Deng et al. (2019), the equation (A7) can be solved using an adaptive SGMCMC algorithm, which works by iterating between the following two steps:

  (a) (*Sampling*) Generate $\boldsymbol{Y}_{\mathrm{mis}}^{(k+1)}$ from a transition kernel induced by a stochastic gradient MCMC algorithm, e.g., stochastic gradient Hamilton Monte Carlo (SGHMC) (Chen et al., 2014).

  (b) (*Parameter updating*) Set $\boldsymbol{\theta}^{(k+1)} = \boldsymbol{\theta}^{(k)} + \gamma_{k+1}g(\boldsymbol{Y}_{\mathrm{mis}}^{(k+1)}, U_{k+1})$, where $\gamma_{k+1}$ denotes the step size used in the stochastic approximation procedure.

The pseudo-code of the adaptive SGHMC algorithm is given by Algorithm 2, where we let $\boldsymbol{\theta}_i = (\boldsymbol{w}_i, \boldsymbol{b}_i)$ denote the parameters associated with the $i$-th layer for $i = 1, 2, \ldots, h+1$, let $(\boldsymbol{Y}_0^{(s,k)}, \boldsymbol{Y}_{h+1}^{(s,k)}) = (\boldsymbol{X}^{(s)}, \boldsymbol{Y}^{(s)})$ denote a training sample $s$, and let $\boldsymbol{Y}_{mis}^{(s,k)} = (\boldsymbol{Y}_1^{(s,k)}, \ldots, \boldsymbol{Y}_h^{(s,k)})$ denote the latent variables imputed for the training sample $s$ at iteration $k$. Occasionally, we use the notation $\boldsymbol{Y}_0^{(s,k)} = \boldsymbol{Y}_0^{(s)} = \boldsymbol{X}^{(s)}$ and $\boldsymbol{Y}_{h+1}^{(s,k)} = \boldsymbol{Y}_{h+1}^{(s)} = \boldsymbol{Y}^{(s)}$.

This algorithm is called "adaptive" as the transition kernel used in step (i) changes with iterations through the working estimate $\boldsymbol{\theta}^{(k)}$. Algorithm 2 is expected to outperform the basic algorithm by Deng et al. (2019), where the stochastic gradient Langevin dynamics (SGLD) algorithm (Welling & Teh, 2011) is used in the sampling step, due to the accelerated convergence of SGHMC over SGLD (Nemeth & Fearnhead, 2019). The convergence of Algorithm 2 is shown in the following lemma, see Section D.2 for the proof.

**Lemma A2.** *Suppose Assumption A7 hold. In Algorithm 2, if we set $\epsilon_k = C_{\epsilon}/(c_e + k^{\alpha})$ and $\gamma_k = C_{\gamma}/(c_g + k^{\alpha})$ for some constants $\alpha \in (0,1)$, $C_{\epsilon} > 0$, $C_{\gamma} > 0$, $c_e \geq 0$ and $c_g \geq 0$, then there exists an iteration $k_0$ and a constant $\lambda_0 > 0$ such that for any $k > k_0$,*

$$\mathbb{E}(\|\hat{\boldsymbol{\theta}}^{(k)} - \widehat{\boldsymbol{\theta}}_n^*\|^2) \leq \lambda_0 \gamma_k, \qquad \text{(A10)}$$

*where $\widehat{\boldsymbol{\theta}}_n^*$ denotes a solution to equation (A7).*

A similar result to Lemma A2 has been established in Liang et al. (2022), except that the penalty term $P_{\lambda}(\boldsymbol{\theta})$ is not included in estimation of $\boldsymbol{\theta}$. As mentioned previously, the adaptive SGHMC algorithm can be more efficient than the IRO algorithm when the training sample size is large. We note that both algorithms can suffer from local traps. To address this issue, a similar procedure as the prior annealing strategy proposed in (Sun et al., 2021) can be used, i.e. start with an over-parametrized model and gradually increase the regularization parameter from 0 to the desired value along with iterations.

## D.2 CONVERGENCE OF ALGORITHM 2

*Notations:* We let $\boldsymbol{D}$ denote a dataset of $n$ observations, and let $D_i$ denote the $i$-th observation of $\boldsymbol{D}$. For StoNet, $D_i$ has included both the input and output variables of the observation. For simplicity of notation, we re-denote the latent variable corresponding to $D_i$ by $Z_i$, and denote by $f_{D_i}(z_i,\boldsymbol{\theta}) = -\log \pi(z_i|D_i,\boldsymbol{\theta})$ the negative log-density function of $Z_i$. Let $\boldsymbol{z} = (z_1, z_2, \ldots, z_n)$ be a realization of $\boldsymbol{Z} = (Z_1, Z_2, \ldots, Z_n)$, and let $F_{\boldsymbol{D}}(\boldsymbol{z},\boldsymbol{\theta}) = \sum_{i=1}^{n} f_{D_i}(z_i,\boldsymbol{\theta})$.

To complete the proof, we need the following assumptions:

**Algorithm 2** An adaptive SGHMC algorithm for training StoNet

**Input**: Dataset $(\boldsymbol{X}, \boldsymbol{Y})$, total iteration number $K$, Monte Carlo step number $t_{HMC}$, the learning rate sequence $\{\epsilon_{k,i} : t = 1, 2, \ldots, T; i = 1, 2, \ldots, h+1\}$, and the step size sequence $\{\gamma_{k,i} : t = 1, 2, \ldots, T; i = 1, 2, \ldots, h+1\}$.

**Initialization**: Randomly initialize the network parameters $\hat{\boldsymbol{\theta}}^{(0)} = (\hat{\boldsymbol{\theta}}_1^{(0)}, \ldots, \hat{\boldsymbol{\theta}}_{h+1}^{(0)})$.

**for** $k = 1$ **to** $K$ **do**

  **STEP 0: Subsampling**: Draw a mini-batch of data and denote it by $S_k$.

  **STEP 1: Backward Sampling**: For each observation $s \in S_k$, sample $\boldsymbol{Y}_i$'s in the order from layer $h$ to layer 1. More explicitly, we sample $\boldsymbol{Y}_i^{(s,k)}$ from the distribution $\pi(\boldsymbol{Y}_i^{(s,k)} | \hat{\theta}_i^{(k-1)}, \hat{\theta}_{i+1}^{(k-1)}, \boldsymbol{Y}_{i+1}^{(s,k)}, \boldsymbol{Y}_{i-1}^{(s,k)}) \propto \pi(\boldsymbol{Y}_{i+1}^{(s,k)} | \hat{\theta}_{i+1}^{(k-1)}, \boldsymbol{Y}_i^{(s,k)}) \pi(\boldsymbol{Y}_i^{(s,k)} | \hat{\theta}_i^{(k-1)}, \boldsymbol{Y}_{i-1}^{(s,k)})$ by running SGHMC for $t_{HMC}$ steps:

  Initialize $\boldsymbol{v}_i^{(s,0)} = \boldsymbol{0}$, and initialize $\boldsymbol{Y}_i^{(s,k,0)}$ by forward pass of DNN.

  **for** $l = 1$ **to** $t_{HMC}$ **do**

    **for** $i = h$ **to** $1$ **do**

      Simulate latent variables

$$
\begin{aligned}
\boldsymbol{v}_i^{(s,k,l)} =&(1 - \epsilon_{k,i}\eta_i)\boldsymbol{v}_i^{(s,k,l-1)} + \epsilon_{k,i}\nabla_{\boldsymbol{Y}_i^{(s,k,l-1)}} \log \pi\left(\boldsymbol{Y}_i^{(s,k,l-1)} \mid \hat{\boldsymbol{\theta}}_i^{(k-1)}, \boldsymbol{Y}_{i-1}^{(s,k,l-1)}\right) \\
&+ \epsilon_{k,i}\nabla_{\boldsymbol{Y}_i^{(s,k,l-1)}} \log \pi\left(\boldsymbol{Y}_{i+1}^{(s,k,l-1)} \mid \hat{\boldsymbol{\theta}}_{i+1}^{(k-1)}, \boldsymbol{Y}_i^{(s,k,l-1)}\right) + \sqrt{2\epsilon_{k,i}\eta}\boldsymbol{e}^{(s,k,l)}, \\
\boldsymbol{Y}_i^{(s,k,l)} =&\boldsymbol{Y}_i^{(s,k,l-1)} + \epsilon_{k,i}\boldsymbol{v}_i^{(s,k,l-1)},
\end{aligned}
\tag{A8}
$$

      where $\boldsymbol{e}^{s,k,l} \sim N(0, \boldsymbol{I}_{d_i})$, $\epsilon_{k,i}$ is the learning rate, and $\eta$ is the friction coefficient. The algorithm is reduced to SGLD when $\epsilon_{k,i}\eta_i \equiv 1$.

    **end for**

  **end for**

  Set $\boldsymbol{Y}_i^{(s,k)} = \boldsymbol{Y}_i^{(s,k,t_{HMC})}$ for $i = 1, 2, \ldots, h$.

  **STEP 2: Parameter Update**: Update the estimates of $\hat{\boldsymbol{\theta}}^{(k-1)} = (\hat{\boldsymbol{\theta}}_1^{(k-1)}, \hat{\boldsymbol{\theta}}_2^{(k-1)}, \ldots, \hat{\boldsymbol{\theta}}_{h+1}^{(k-1)})$ by stochastic gradient descent

$$
\hat{\boldsymbol{\theta}}_i^{(k)} = \hat{\boldsymbol{\theta}}_i^{(k-1)} + \gamma_{k,i}\left(\frac{n}{|S_k|}\sum_{s \in S_k} \nabla_{\boldsymbol{\theta}_i} \log \pi(Y_i^{(s,k)}|\hat{\boldsymbol{\theta}}_i^{(k-1)}, Y_{i-1}^{(s,k)}) - n\nabla_{\boldsymbol{\theta}_i} P_\lambda(\hat{\boldsymbol{\theta}}_i)\right), \tag{A9}
$$

  for $i = 1, 2, \ldots, h+1$, where $\gamma_{k,i}$ is the step size used for updating $\theta_i$.

**end for**

**Output**: $\hat{\theta}_n^{(K)}$.

**Assumption A7.** *(i) The function $F_{\boldsymbol{D}}(\cdot,\cdot)$ takes nonnegative real values, and there exist constants $A, B \geq 0$, such that $|F_{\boldsymbol{D}}(\boldsymbol{0},\boldsymbol{\theta}^*)| \leq \mathbf{A}$, $\|\nabla_{\boldsymbol{Z}} F_{\boldsymbol{D}}(\boldsymbol{0},\boldsymbol{\theta}^*)\| \leq \mathbf{B}$, $\|\nabla_{\boldsymbol{\theta}} F_{\boldsymbol{D}}(\boldsymbol{0},\boldsymbol{\theta}^*)\| \leq \mathbf{B}$, and $\|H(\boldsymbol{0},\boldsymbol{\theta}^*)\| \leq \mathbf{B}$.*

*(ii) (Smoothness) $F_{\boldsymbol{D}}(\cdot,\cdot)$ is $M$-smooth and $H(\cdot,\cdot)$ is $M$-Lipschitz: there exists some constant $M > 0$ such that for any $\boldsymbol{Z}, \boldsymbol{Z}' \in \mathbb{R}^{d_z}$ and any $\boldsymbol{\theta}, \boldsymbol{\theta}' \in \Theta$,*

$$\|\nabla_{\boldsymbol{Z}} F_{\boldsymbol{D}}(\boldsymbol{Z},\boldsymbol{\theta}) - \nabla_{\boldsymbol{Z}} F_{\boldsymbol{D}}(\boldsymbol{Z}',\boldsymbol{\theta}')\| \leq M\|\boldsymbol{Z} - \boldsymbol{Z}'\| + M\|\boldsymbol{\theta} - \boldsymbol{\theta}'\|,$$
$$\|\nabla_{\boldsymbol{\theta}} F_{\boldsymbol{D}}(\boldsymbol{Z},\boldsymbol{\theta}) - \nabla_{\boldsymbol{\theta}} F_{\boldsymbol{D}}(\boldsymbol{Z}',\boldsymbol{\theta}')\| \leq M\|\boldsymbol{Z} - \boldsymbol{Z}'\| + M\|\boldsymbol{\theta} - \boldsymbol{\theta}'\|,$$
$$\|H(\boldsymbol{Z},\boldsymbol{\theta}) - H(\boldsymbol{Z}',\boldsymbol{\theta}')\| \leq M\|\boldsymbol{Z} - \boldsymbol{Z}'\| + M\|\boldsymbol{\theta} - \boldsymbol{\theta}'\|.$$

*(iii) (Dissipativity) For any $\boldsymbol{\theta} \in \Theta$, the function $F_{\boldsymbol{D}}(\cdot,\boldsymbol{\theta}^*)$ is $(m,b)$-dissipative: there exist some constants $m > \frac{1}{2}$ and $b \geq 0$ such that $\langle \boldsymbol{Z}, \nabla_{\boldsymbol{Z}} F_{\boldsymbol{D}}(\boldsymbol{Z},\boldsymbol{\theta}^*)\rangle \geq m\|\boldsymbol{Z}\|^2 - b$.*

*(iv) (Gradient noise) There exists a constant $\varsigma \in [0,1)$ such that for any $\boldsymbol{Z}$ and $\boldsymbol{\theta}$, $\mathbb{E}\|\nabla_{\boldsymbol{Z}} \hat{F}_{\boldsymbol{D}}(\boldsymbol{Z},\boldsymbol{\theta}) - \nabla_{\boldsymbol{Z}} F_{\boldsymbol{D}}(\boldsymbol{Z},\boldsymbol{\theta})\|^2 \leq 2\varsigma(M^2\|\boldsymbol{Z}\|^2 + M^2\|\boldsymbol{\theta} - \boldsymbol{\theta}^*\|^2 + B^2)$.*

*(v) The step size $\{\gamma_k\}_{k\in\mathbb{N}}$ is a positive decreasing sequence such that $\gamma_k \to 0$ and $\sum_{k=1}^{\infty} \gamma_k = \infty$. In addition, let $h(\boldsymbol{\theta}) = \mathbb{E}(H(\boldsymbol{Z},\boldsymbol{\theta}))$, then there exists $\delta > 0$ such that for any $\boldsymbol{\theta} \in \Theta$, $\langle \boldsymbol{\theta} - \boldsymbol{\theta}^*, h(\boldsymbol{\theta}))\rangle \geq \delta\|\boldsymbol{\theta} - \boldsymbol{\theta}^*\|^2$, and $\liminf_{k\to\infty} 2\delta\frac{\gamma_k}{\gamma_{k+1}} + \frac{\gamma_{k+1} - \gamma_k}{\gamma_{k+1}^2} > 0$.*

*(vi) (Solution of Poisson equation) For any $\boldsymbol{\theta} \in \Theta$, $\boldsymbol{z} \in \mathfrak{Z}$, and a function $V(\boldsymbol{z}) = 1 + \|\boldsymbol{z}\|$, there exists a function $\mu_{\boldsymbol{\theta}}$ on $\mathfrak{Z}$ that solves the Poisson equation $\mu_{\boldsymbol{\theta}}(\boldsymbol{z}) - \mathcal{T}_{\boldsymbol{\theta}}\mu_{\boldsymbol{\theta}}(\boldsymbol{z}) = H(\boldsymbol{\theta},\boldsymbol{z}) - h(\boldsymbol{\theta})$, where $\mathcal{T}_{\boldsymbol{\theta}}$ denotes a probability transition kernel with $\mathcal{T}_{\boldsymbol{\theta}}\mu_{\boldsymbol{\theta}}(\boldsymbol{z}) = \int_{\mathfrak{Z}} \mu_{\boldsymbol{\theta}}(\boldsymbol{z}')\mathcal{T}_{\boldsymbol{\theta}}(\boldsymbol{z},\boldsymbol{z}')d\boldsymbol{z}'$, such that*

$$H(\boldsymbol{\theta}_k,\boldsymbol{z}_{k+1}) = h(\boldsymbol{\theta}_k) + \mu_{\boldsymbol{\theta}_k}(\boldsymbol{z}_{k+1}) - \mathcal{T}_{\boldsymbol{\theta}_k}\mu_{\boldsymbol{\theta}_k}(\boldsymbol{z}_{k+1}), \quad k = 1, 2, \ldots. \tag{A11}$$

*Moreover, for all $\boldsymbol{\theta}, \boldsymbol{\theta}' \in \Theta$ and $\boldsymbol{z} \in \mathfrak{Z}$, we have $\|\mu_{\boldsymbol{\theta}}(\boldsymbol{z}) - \mu_{\boldsymbol{\theta}'}(\boldsymbol{z})\| \leq \varsigma_1\|\boldsymbol{\theta} - \boldsymbol{\theta}'\|V(\boldsymbol{z})$ and $\|\mu_{\boldsymbol{\theta}}(\boldsymbol{z})\| \leq \varsigma_2 V(\boldsymbol{z})$ for some constants $\varsigma_1 > 0$ and $\varsigma_2 > 0$.*

The smoothness and dissipativity conditions are regular for studying the convergence of stochastic gradient MCMC algorithms, and they have been used in many papers such as Raginsky et al. (2017) and Gao et al. (2021). As implied by the definition of $F_{\boldsymbol{D}}(\boldsymbol{z},\boldsymbol{\theta})$, the values of $M$, $m$ and $b$ increase linearly with the sample size $n$. Therefore, we can impose a nonzero lower bound on $m$ to facilitate related proofs.

Assumption A7-(iv) introduces an extra constant $\varsigma$ to facilitate our study. For the full data case, we have $\varsigma = 0$, i.e., the gradient $\nabla_{\boldsymbol{Z}} F_{\boldsymbol{D}}(\boldsymbol{Z},\boldsymbol{\theta})$ can be evaluated accurately.

As shown by Benveniste et al. (1990) (p.244), Assumption A7-(v) can be satisfied by setting $\gamma_k = \tilde{a}/(\tilde{b} + k^\alpha)$ for some constants $\tilde{a} > 0$, $\tilde{b} \geq 0$, and $\alpha \in (0, 1 \wedge 2\delta\tilde{a})$. By (A9), $\delta$ increases linearly with the sample size $n$. Therefore, if we set $\tilde{a} = \Omega(1/n)$ then $2\delta\tilde{a} > 1$ can be satisfied, where $\Omega(\cdot)$ denotes the order of the lower bound of a function. In this paper, we simply choose $\alpha \in (0,1)$ by assuming that $\tilde{a}$ has been set appropriately with $2\delta\tilde{a} \geq 1$ held.

Assumption A7-(vi) is also regular for studying the convergence of stochastic gradient MCMC algorithms, see e.g., Whye et al. (2016) and Deng et al. (2019). Alternatively, one can assume that the MCMC algorithms satisfy the drift condition, and then Assumption A7-(vi) can be verified, see e.g., Andrieu et al. (2005).

**Outline of the proof of Lemma A2** Lemma A2 can be proved in a similar way to Theorem 1 of Deng et al. (2019). However, since Algorithm 2 employs SGHMC for updating $\boldsymbol{Z}^{(k)}$, which is mathematically different from the SGLD algorithm employed in Deng et al. (2019), Lemma 1 of Deng et al. (2019) (uniform $L_2$ bounds of $\boldsymbol{\theta}^{(k)}$ and $\boldsymbol{Z}^{(k)}$) cannot be applied any more. A similar result to Lemma 1 of Deng et al. (2019) has been established in Liang et al. (2022) under appropriate conditions of $\{\epsilon_k\}$ and $\{\gamma_k\}$ as prescribed in this lemma, where it is shown that $\mathbb{E}\|\boldsymbol{\theta}^{(k)}\|^2 \leq C_{\boldsymbol{\theta}}$, $\mathbb{E}\|\boldsymbol{V}^{(k)}\|^2 \leq C_{\boldsymbol{V}}$ and $\mathbb{E}\|\boldsymbol{Z}^{(k)}\|^2 \leq C_{\boldsymbol{Z}}$ for some constants $C_{\boldsymbol{\theta}}$, $C_{\boldsymbol{V}}$ and $C_{\boldsymbol{Z}}$.

Note that in the proof of Lemma A2, the boundedness of $\Theta$ is not assumed. In Liang et al. (2022), an explicit expression of $\lambda_0$ has been given. For simplicity, we have the expression omitted in this paper.

# E   Prediction Uncertainty Quantification for Sparse StoNet

Let's first consider the case that we have a regression StoNet trained by the IRO algorithm. In this case, the prediction uncertainty can be quantified by a recursive application of Eve's law. Extension of the results to other cases will be discussed later.

Let $\boldsymbol{z}$ denote a test point at which the prediction uncertainty is to be quantified. For simplicity of notations, we suppress the bias term by including it as a special column of the corresponding weight matrix. To indicate the iterative nature of the training algorithm, we include the superscript 't' in the derivation. Let $\boldsymbol{Z}_i^{(t)}$ denote the imputed latent variable at layer $i$, corresponding to the input vector $\boldsymbol{z}$. For convenience, we let $\boldsymbol{Z}_0^{(t)} = \boldsymbol{z}$ for all $t$. Let $\boldsymbol{\mu}_i^{(t)}$ and $\boldsymbol{\Sigma}_i^{(t)}$ denote, respectively, the mean and covariance matrix of $\boldsymbol{Z}_i^{(t)}$. Let $\boldsymbol{w}_{i_j}^{(t)}$ denote the $j$-th row of the weight matrix $\boldsymbol{w}_i^{(t)}$, which represents the weights from the neurons of layer $i-1$ to neuron $j$ of layer $i$. By Eve's law, for any layer $i \in \{2, 3, \ldots, h+1\}$,

$$
\begin{aligned}
\boldsymbol{\Sigma}_i^{(t)} &= \mathbb{E}(\text{Var}(\boldsymbol{Z}_i^{(t)}|\boldsymbol{Z}_{i-1}^{(t)})) + \text{Var}(\mathbb{E}(\boldsymbol{Z}_i^{(t)}|\boldsymbol{Z}_{i-1}^{(t)})) \\
&= \mathbb{E}\text{diag}\Big\{\psi(\boldsymbol{Z}_{i-1}^{(t)}))^T \text{Var}(\hat{\boldsymbol{w}}_{i_j}^{(t)})\psi(\boldsymbol{Z}_{i-1}^{(t)})) : j = 1, \ldots d_i\Big\} + \text{Var}(\mathbb{E}(\hat{\boldsymbol{w}}_i)\psi(\boldsymbol{Z}_{i-1}^{(t)})) \\
&= \text{diag}\Big\{ tr(\text{Var}(\hat{\boldsymbol{w}}_{i_j}^{(t)}))\text{Var}(\psi(\boldsymbol{Z}_{i-1}^{(t)}))) + (\mathbb{E}(\psi(\boldsymbol{Z}_{i-1}^{(t)})))^T \\
&\quad \times \text{Var}(\hat{\boldsymbol{w}}_{i_j}^{(t)})(\mathbb{E}(\psi(\boldsymbol{Z}_{i-1}^{(t)}))) : j = 1, \ldots d_i\Big\} + \mathbb{E}(\hat{\boldsymbol{w}}_i)\text{Var}(\psi(\boldsymbol{Z}_{i-1}^{(t)}))(\mathbb{E}(\hat{\boldsymbol{w}}_i))^T,
\end{aligned}
$$

where $\text{Var}(\hat{\boldsymbol{w}}_{i_j}^{(t)})$ is calculated by the Lasso+OLS or Lasso+mLS procedure suggested by Liu & Yu (2013). That is, we estimate $\boldsymbol{w}_{i_j}^{(t)}$ by applying the ordinary least square (OLS) or a modified least square (mLS) procedure to the regression model selected by Lasso. We refer to Theorem 3 of Liu & Yu (2013) for asymptotic normality of the non-sparse components of $\hat{\boldsymbol{w}}_{i_j}^{(t)}$. For the OLS case, the non-sparse submatrix of $\text{Var}(\hat{\boldsymbol{w}}_{i_j}^{(t)})$ is given by

$$
\widetilde{\text{Var}(\hat{\boldsymbol{w}}_{i_j}^{(t)})} = \hat{\varsigma}_{i,j}^2[(\psi(\widetilde{\mathbb{Y}}_{i-1}^{(t)})^T\psi(\widetilde{\mathbb{Y}}_{i-1}^{(t)})]^{-1},
$$

where $\psi(\widetilde{\mathbb{Y}}_{i-1}^{(t)})$ denotes the design matrix of the linear regression $\mathbb{Y}_{i,j}^{(t)} = \psi(\widetilde{\mathbb{Y}}_{i-1}^{(t)})(\tilde{\boldsymbol{w}}_{i_j}^{(t)})^T + \boldsymbol{\epsilon}_{i,j}$ selected by Lasso for neuron $j$ of layer $i$ at iteration $t$, $\boldsymbol{\epsilon}_{i,j} \sim N(0, \varsigma_i^2 I_n)$, and $\hat{\varsigma}_{i,j}^2$ denotes the OLS estimator of $\varsigma_i^2$. Here $\mathbb{Y}_{i-1}^{(t)} \in \mathbb{R}^{n \times d_{i-1}}$ denotes the imputed latent variables for all neurons of layer $i-1$, $\mathbb{Y}_{i,j}^{(t)} \in \mathbb{R}^n$ denotes the imputed latent variables for neuron $j$ of layer $i$, $\widetilde{\mathbb{Y}}_{i-1}^{(t)} \in \mathbb{R}^{n \times \tilde{q}_{i,j}}$ denotes the variables selected by Lasso, $\tilde{\boldsymbol{w}}_{i_j}^{(t)}$ denotes the corresponding regression coefficients, and $\tilde{q}_{i,j}$ denotes the number of selected variables.

Let $\boldsymbol{\mu}_{i-1}^{(t)} = (\mu_{i-1,1}^{(t)}, \ldots, \mu_{i-1,d_{i-1}}^{(t)})^T$ denote the mean of $\boldsymbol{Z}_{i-1}^{(t)}$, and let $D_{\psi'}(\boldsymbol{\mu}_{i-1}^{(t)}) = \text{diag}\{\psi'(\mu_{i-1,1}^{(t)}), \ldots, \psi'(\mu_{i-1,d_{i-1}}^{(t)})\}$, where $\psi'$ denotes the first derivative of the activation function $\psi$. By the first order Taylor expansion, we have

$$
\mathbb{E}(\psi(\boldsymbol{Z}_{i-1}^{(t)})) \approx \psi(\boldsymbol{\mu}_{i-1}^{(t)}),
$$
$$
\text{Var}(\psi(\boldsymbol{Z}_{i-1}^{(t)})) \approx D_{\psi'}(\boldsymbol{\mu}_{i-1}^{(t)})\Sigma_{i-1}^{(t)}D_{\psi'}(\boldsymbol{\mu}_{i-1}^{(t)}).
$$

Further, if we estimate $\mathbb{E}(\hat{\boldsymbol{w}}_i)$ by $\hat{\boldsymbol{w}}_i$ and estimate $\boldsymbol{\mu}_{i-1}^{(t)}$ by $\boldsymbol{Z}_{i-1}^{(t)}$, then we have the approximation:

$$
\begin{aligned}
\widehat{\boldsymbol{\Sigma}}_i^{(t)} \approx \text{diag}\Big\{ & tr\big(\text{Var}(\hat{\boldsymbol{w}}_{i_j}^{(t)})D_{\psi'}(\boldsymbol{Z}_{i-1}^{(t)})\widehat{\Sigma}_{i-1}^{(t)}D_{\psi'}(\boldsymbol{Z}_{i-1}^{(t)})\big) \\
& + (\psi(\boldsymbol{Z}_{i-1}^{(t)}))^T\text{Var}(\hat{\boldsymbol{w}}_{i_j}^{(t)})\psi(\boldsymbol{Z}_{i-1}^{(t)}) : j = 1, \ldots, d_i\Big\} \\
& + \hat{\boldsymbol{w}}_i^{(t)}D_{\psi'}(\boldsymbol{Z}_{i-1}^{(t)})\widehat{\Sigma}_{i-1}^{(t)}D_{\psi'}(\boldsymbol{Z}_{i-1}^{(t)})(\hat{\boldsymbol{w}}_i^{(t)})^T.
\end{aligned}
\tag{A12}
$$

For the first hidden layer, it is reduced to

$$
\widehat{\boldsymbol{\Sigma}}_1^{(t)} \approx \text{diag}\Big\{tr\big(\text{Var}(\hat{\boldsymbol{w}}_{1_j}^{(t)})\text{Var}(\boldsymbol{z})\big) + \boldsymbol{z}^T\text{Var}(\hat{\boldsymbol{w}}_{1_j}^{(t)})\boldsymbol{z} : j = 1, \ldots, d_1\Big\} + \hat{\boldsymbol{w}}_1^{(t)}\text{Var}(\boldsymbol{z})(\hat{\boldsymbol{w}}_1^{(t)})^T.
\tag{A13}
$$

Since $\text{Var}(\boldsymbol{z}) = 0$ holds for the fixed test point $\boldsymbol{z}$, $\widehat{\boldsymbol{\Sigma}}_1^{(t)}$ can be further reduced to

$$\widehat{\boldsymbol{\Sigma}}_1^{(t)} \approx \text{diag}\left\{\boldsymbol{z}^T \text{Var}(\hat{\boldsymbol{w}}_{i_j}^{(t)})\boldsymbol{z} : j = 1, 2, \ldots, d_1\right\}.$$

Let $\mu(\boldsymbol{z}, \hat{\boldsymbol{\theta}})$ denote the prediction of a StoNet with weights $\hat{\boldsymbol{\theta}}$ at point $\boldsymbol{z}$. Note that the StoNet (2) has the same prediction function as the DNN (1), i.e., the random noise added to the latent variables is set to 0 in forward prediction. Suppose that a set of StoNet estimates, $\mathcal{S} = \{\hat{\boldsymbol{\theta}}^{(1)}, \hat{\boldsymbol{\theta}}^{(2)}, \ldots, \hat{\boldsymbol{\theta}}^{(m)}\}$, has been collected after convergence of the IRO algorithm. Given $\widehat{\Sigma}_i^{(t)}$'s, by the Wald method, the 95% prediction interval of $\mu_j(\boldsymbol{z}, \boldsymbol{\theta}^*)$, the $j$-th component of $\mu(\boldsymbol{z}, \boldsymbol{\theta}^*)$, can be constructed as described in Section 4 of the main text.

The proposed confidence interval construction procedure can be easily extended to the StoNet with a logistic regression output layer via the Wald/endpoint transformation. Following from Lemma A2, the proposed procedure can also be applied to the StoNet trained by the adaptive stochastic gradient MCMC algorithm.

## F  SUPPLEMENTARY NUMERICAL RESULTS

### F.1  SUPPLEMENTARY RESULTS FOR SECTION 5

Figure A1 shows the variable selection path for the model (8), and Figure A2 shows the prediction intervals produced by the StoNet on some test points.

(a) 1-hidden-layer StoNet          (b) 1-hidden-layer DNN

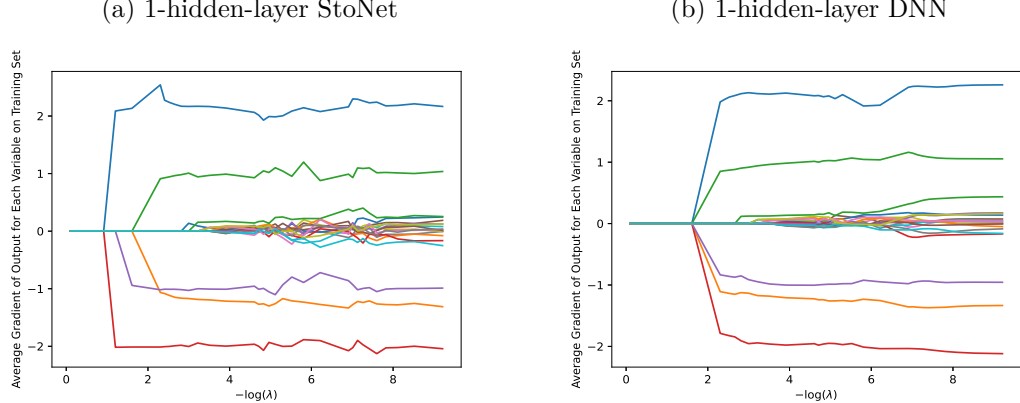

Figure A1: Variable selection paths by the StoNet (under the single-$\sigma^2$ setting) and DNN for the model (8), where $y$-axis is the average output gradient $\frac{1}{n}\sum_{k=1}^{n} \frac{\partial \hat{\mu}(\boldsymbol{x})}{\partial x_i}|_{\boldsymbol{x}^{(k)}}$ calculated over the training data, and $x$-axis is $-\log(\lambda)$.

## G  HYPER-PARAMETER SETTING

For the StoNet, since the learning rates $\epsilon_{k,i}$'s and the latent variable variances $\sigma_i^2$'s can be largely canceled at each step of latent variable imputation, their absolute values do not mean much to the convergence of Algorithm 2. For this reason, we often set their values to be very small in our numerical experiments, which merely controls the size of random noise added to the corresponding latent variables.

### G.1  AN ILLUSTRATIVE EXAMPLE

**One-hidden-layer StoNet:**  We tried three parameter settings:

(a) $\sigma_2^2 = 0.5e - 4$, $\sigma_1^2 = 0.5e - 5$, $\epsilon_{k,1} = 0.5e - 8$, $\eta_i = \frac{1}{\epsilon_{k,i}}$, $t_{HMC} = 1$, $\frac{\gamma_{k,1}}{|S_k|} = 5e - 4$, $\frac{\gamma_{k,2}}{|S_k|} = 5e - 8$, $|S_k| = 50$;

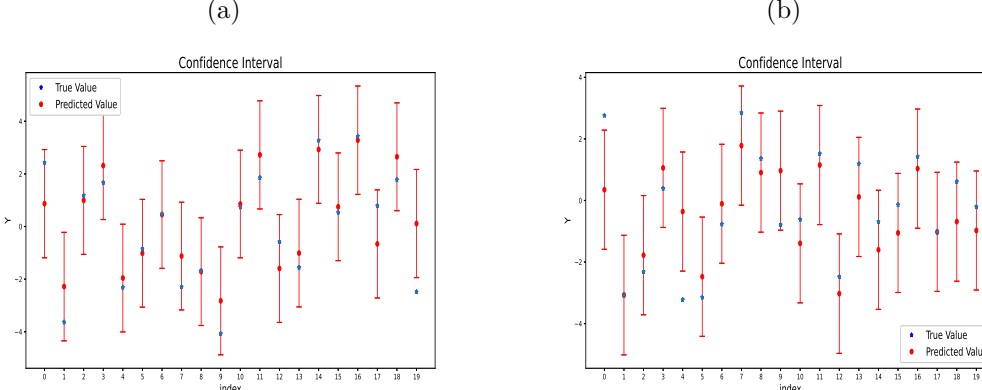

Figure A2: Prediction Intervals produced by (a) one-hidden-layer StoNet and (b) two-hidden-layer StoNet at 20 test points, where the StoNets were trained under the single-$\sigma^2$ setting.

(b) $\sigma_2^2 = 1e-4$, $\sigma_1^2 = 1e-5$, $\epsilon_{k,1} = 1e-8$, $\eta_i = \frac{1}{\epsilon_{k,i}}$, $t_{HMC} = 1$, $\frac{\gamma_{k,1}}{|S_k|} = 5e-4$, $\frac{\gamma_{k,2}}{|S_k|} = 5e-8$, $|S_k| = 50$;

(c) $\sigma_2^2 = 2e-4$, $\sigma_1^2 = 2e-5$, $\epsilon_{k,1} = 2e-8$, $\eta_i = \frac{1}{\epsilon_{k,i}}$, $t_{HMC} = 1$, $\frac{\gamma_{k,1}}{|S_k|} = 5e-4$, $\frac{\gamma_{k,2}}{|S_k|} = 5e-8$, $|S_k| = 50$.

**Two-hidden-layer StoNet:** We tried three parameter settings:

(a) $\sigma_3^2 = 0.5e-9, \sigma_2^2 = 0.5e-10, \sigma_1^2 = 0.5e-11$, $\epsilon_{k,2} = 0.5e-13, \epsilon_{k,1} = 1e-14$, $\eta_i = \frac{1}{\epsilon_{k,i}}$, $t_{HMC} = 1$, $\frac{\gamma_{k,3}}{|S_k|} = 5e-6$, $\frac{\gamma_{k,2}}{|S_k|} = 5e-10$, $\frac{\gamma_{k,1}}{|S_k|} = 5e-14$, $|S_k| = 50$;

(b) $\sigma_3^2 = 1e-9, \sigma_2^2 = 1e-10, \sigma_1^2 = 1e-11$, $\epsilon_{k,2} = 1e-13, \epsilon_{k,1} = 1e-14$, $\eta_i = \frac{1}{\epsilon_{k,i}}$, $t_{HMC} = 1$, $\frac{\gamma_{k,3}}{|S_k|} = 5e-6$, $\frac{\gamma_{k,2}}{|S_k|} = 5e-10$, $\frac{\gamma_{k,1}}{|S_k|} = 5e-14$, $|S_k| = 50$;

(c) $\sigma_3^2 = 2e-9, \sigma_2^2 = 2e-10, \sigma_1^2 = 2e-11$, $\epsilon_{k,2} = 1e-13, \epsilon_{k,1} = 1e-14$, $\eta_i = \frac{1}{\epsilon_{k,i}}$, $t_{HMC} = 1$, $\frac{\gamma_{k,3}}{|S_k|} = 5e-6$, $\frac{\gamma_{k,2}}{|S_k|} = 5e-10$, $\frac{\gamma_{k,1}}{|S_k|} = 5e-14$, $|S_k| = 50$.

For both StoNets, the major differences of the settings is at $\sigma_i$'s. For convenience, we call the settings (a), (b) and (c) by half-$\sigma^2$ setting, single-$\sigma^2$ setting, and double-$\sigma^2$ setting, respectively.

## G.2 CoverType Data

We used 80% of the data for training and other 20% data for testing. We fit the data by a DNN of two hidden layers, with 1000 and 500 hidden units, respectively. The DNN was trained for 2000 epochs using SGD with momentum, a constant learning rate of 0.01, a momentum coefficient of 0.9, and a batch size of 500. Different regularization parameters were tried with $\lambda$ ranging from $1e-6$ to $5e-3$.

## G.3 CIFAR10

We conduct experiments on CIFAR10 data sets. Following the setting of post-calibration methods in Guo et al. (2017), we split the training data into a training set of 45000 images and a hold out validation set of 5000 images for calibration. The training settings for 3 models are:

- ResNet110: Model is trained on training set with SGD with momentum for 200 epochs with batch size 128, momentum 0.9 and weight decay 0.0001. Learning rate was set to 0.1 for first 80 epochs and divided by 10 at 80-th and 150-th epoch.

- Densenet40: Model is trained on training set with SGD with momentum for 300 epochs with batch size 128, momentum 0.9 and weight decay 0.0001. Learning rate was set to 0.1 for first 150 epochs and divided by 10 at 150-th and 225-th epoch.
- WideResNet-28-10: Model is trained on training set with SGD with momentum for 200 epochs with momentum 0.9 and weight decay 0.0005. Learning rate was set to 0.1 for first 60 epochs and divided by 10 at 60-th, 120-th epoch and 160-th epoch.

After training, we compute the input of the last fully connected layer of each model on the validation set, and use them as input to a StoNet model, we refit a StoNet model with 1 hidden layer, 100 hidden units and tanh as activation function. The StoNet model is trained with algorithm (2) with hyper-parameters given in table A1.

| Hyper-Parameter | Value |
|---|---|
| $[\sigma_1^2, \sigma_2^2]$ | [1e-2, 1e-3] |
| $\epsilon_{k,1}$ | 1e-7 |
| $\eta_1$ | $\frac{1}{\epsilon_{k,1}}$ |
| $t_{HMC}$ | 1 |
| $[\gamma_{k,1}, \gamma_{k,2}]$ | $[\frac{5e-4}{5000}, \frac{5e-6}{5000}]$ |
| $|S_k|$ | 50 |
| $P_\lambda(\boldsymbol{\theta})$ | $1e-4 \times \|\boldsymbol{\theta}\|_1$ |

Table A1: Post-StoNet Hyper-Parameter Setting for CIFAR10 data

| Hyper-Parameter | Value |
|---|---|
| $[\sigma_1^2, \sigma_2^2]$ | [1e-4, 1e-5] |
| $\epsilon_{k,1}$ | 1e-7 |
| $\eta_1$ | $\frac{1}{\epsilon_{k,1}}$ |
| $t_{HMC}$ | 1 |
| $[\gamma_{k,1}, \gamma_{k,2}]$ | $[\frac{1e-3}{N}, \frac{1e-5}{N}]$ |
| $|S_k|$ | 50 |
| $P_\lambda(\boldsymbol{\theta})$ | $\lambda\|\boldsymbol{\theta}\|_1$ |

Table A2: StoNet Hyper-Parameter Setting for UCI data sets, where $N$ is size of the calibration data set.

### G.4 Regression Examples

The Wine [1], Power Plant[2], Protein[3] and Year[4] data sets are from UCI machine learning repository. For all experiments, we split the data into 40% as training set, 40% as calibration set(used to fit a StoNet model for our methods and used to compute absolute value of residue as non-conformity score for Split Conformal) and 20% as test set. The random split was repeated 20 times and we report the mean and standard deviation of confidence interval length and coverage rate. For training, we use a DNN model with 2 hidden layers, 1000 and 100 hidden units respectively and tanh activation function. The model is trained using Adam(Kingma & Ba, 2015) with batch size 50, constant learning rate 0.001. The model is trained for 5000 epochs for Wine and Power Plant data sets, 1000 Epochs for Protein data set, and 200 epochs for Year data set. After the model is trained, we refit a StoNet on the calibration data set, with output of the last hidden layer of DNN as input. We use a StoNet with one hidden layer, 20 hidden units with *tanh* activation function. Algorithm 2 is used to train the model with hyper-parameters given in table A2. The penalty parameter $\lambda$ is

---

[1]https://archive.ics.uci.edu/dataset/186/wine+quality
[2]https://archive.ics.uci.edu/dataset/294/combined+cycle+power+plant
[3]https://archive.ics.uci.edu/dataset/265/physicochemical+properties+of+protein+tertiary+structure
[4]http://archive.ics.uci.edu/dataset/203/yearpredictionmsd

selected from $\{5e-2, 2e-2, 1e-2, 5e-3, 2e-3, 1e-3, 5e-4\}$ by 5 fold cross validation, where we pick the $\lambda$ such that the average coverage rate on the validation sets are closest to the target level 90%. Specifically, we pick $\lambda = 2e-2$ for Wine data set, $\lambda = 5e-3$ for Power Plant data set, $\lambda = 2e-3$ for Protein data set, $\lambda = 1e-3$ for Year data set.

