# OpenReview forum: "Statistical Inference for Deep Learning via Stochastic Modeling"
_ICLR.cc/2024/Conference — Submitted to ICLR 2024_

### Official Review · Reviewer_GYhy · 2023-10-30

**Soundness:** 2 fair
**Presentation:** 2 fair
**Contribution:** 2 fair
**Rating:** 5
**Confidence:** 3

**Summary:**

In this paper, authors show how the sparse learning theory with Lasso penalty can be adapted to deep neural networks (StoNet) from linear models, and provide a recursive method named post-StoNet to quantify the prediction uncertainty for StoNet.

The numerical results suggest that the StoNet significantly improves prediction uncertainty quantification for deep learning models compared to the conformal method and other post processing calibration methods.

**Strengths:**

1. The work is a combination of existing methods including StoNet, IRO, ASGMCMC and Lasso penalty.
2. The background introduction, problem definition and theoretical derivation are well-described.

**Weaknesses:**

1. The main issues of this paper including **novelty and soundness of the results**.
2. Abstract description is not clear, and writing needs to be improved.
3. Comparision with other method is too few and experiment performance improvement is marginal.

**Questions:**

1. In Figure 2, there is slight difference with regard to overall distribution between StoNet and DNN. The variance of StoNet is larger than DNN, which seems like that vanilla DNN performs better than StoNet. The result is confusing.
2. Figure 3 lacks legend, what does lines of different color represent?
3. In Table 2, the ACC results (mean and std.) of 'No Post Calibration' method and 'Temp. Scaling' method are exactly the same, which is counter-intuitive and is of low probability, what is the reason.
4. As for result in Table 3, the compared method (Vovk et al., 2005) is proposed too long ago. To be more convincing, further ablation study should be done, for example, compare with vanilla StoNet or other related works.
5. In Table 1 and 2 of Liang et al. (2022), results of massive datasets and methods are listed, which is not discussed in this paper.

---

> ### Author Response · Authors · 2023-11-20
> **Rebuttal by Authors**
>
> Thank you for your constructive reviews and comments, please find our point-by-point response below
>
> > **R3W1** The main issues of this paper including novelty and soundness of the results.
>
> The work by Liang et al. (2022)[2] discussed the application of the StoNet in nonlinear sufficient dimension reduction. However, the major novelty of this paper are (i) to develop a sparse StoNet framework, showing that it can serve as a bridge for transferring the theory and methods developed for linear models to deep learning models; (ii) to develop a post-StoNet framework, showing that it provides a general method for faithfully quantifying the prediction uncertainty of large-scale deep learning models.
>
> Furthermore, we have provided rigorous theoretical justifications for the proposed frameworks and illustrated their performance using numerical examples.
>
> > **R3W2** Abstract description is not clear, and writing needs to be improved
>
> Thank you for your comment. We will enhance the abstract by precisely highlighting the contributions of this paper.
>
> > **R3W3** Comparison with other method is too few and experiment performance improvement is marginal.
>
> Thank you for your comment. In the rebuttal, we have conducted more numerical experiments, including a large-scale model for CIFAR 100, an ablation study for the regularization
> parameter $\lambda$, comparison with split Conformalized quantile regression(CQR),
> and one more simulation example for high-dimensional nonlinear variable selection. Please refer to our reply to Reviewer 9U3v and Reviewer HASS for the details (R1W2, R2C3-1, R2C3-3). And in our results, we have included the mean and standard deviation of the metric, which shows that our improvements are significant(see R1W2 of our rebuttal to reviewer 9U3v)
>
> > **R3Q1** In Figure 2, there is slight difference with regard to overall distribution between StoNet and DNN. The variance of StoNet is larger than DNN, which seems like that vanilla DNN performs better than StoNet. The result is confusing.
>
> In Lemma 1, it is shown that the StoNet and DNN are asymptotically equivalent when the noise $\sigma_i$ at each layer goes to 0. The goal of our theoretical results (Theorem 1 and Corollary 1) is to use StoNet as a tool to provide theoretical justification for the consistency of penalized DNN models. Therefore,  for the simulation example of Section 5, it is expected that the DNN and StoNet will produce similar results. Since StoNet introduces more noise during training, it is also expected that the lines in Figure 2(a) have a larger variability than those in Figure 2(b).
> Figure 2 matches with our theory.
>
> > **R3Q2** Figure 3 lacks legend, what does lines of different color represent?
>
> In this figure, each line/color corresponds to a different feature,
> which shows how the gradient of the feature changes with the regularization parameter $\lambda$.
> Our major goal is to use this figure to illustrate that
> the penalized DNN model can identify important features.
>
> > **R3Q3** In Table 2, the ACC results (mean and std.) of 'No Post Calibration' method and 'Temp. Scaling' method are exactly the same, which is counter-intuitive and is of low probability, what is the reason.
>
> The temperature scaling method (Guo et al. 2017 [1]) divides the pre-softmax output of the DNN model by a temperature, and then learns the optimal temperature in a separate calibration set. To be more precise,
> let $(z_1, \dots, z_K)$ be the neural network output for an $K$-class classification problem, and the temperature scaling method will produce $softmax(z_1 / T, z_2 / T, \dots, z_n / T)$ as the predicting probability vector. Since rescaling the values of $z_i$'s doesn't change their order,
> the prediction label
> $$
> \arg\max  softmax(z_1 / T, z_2 / T, \dots, z_n / T),
> $$
> will be the same as the unscaled one. Therefore, temperature scaling won't change prediction labels, and  its prediction accuracy is exactly the same as the original model.
>
>
>
> ### Reference
> [1] Guo, Chuan, et al. "On calibration of modern neural networks." International conference on machine learning. PMLR, 2017.
>
>   [2] Liang, Siqi, Yan Sun, and Faming Liang. "Nonlinear Sufficient Dimension Reduction with a Stochastic Neural Network." Advances in Neural Information Processing Systems 35 (2022): 27360-27373.

---

> ### Author Response · Authors · 2023-11-20
> **Rebuttal by Authors**
>
> > **R3Q4** As for result in Table 3, the compared method (Vovk et al., 2005) is proposed too long ago. To be more convincing, further ablation study should be done, for example, compare with vanilla StoNet or other related works.
>
> The conformal prediction method has become popular in recent years as a general tool to apply on top of almost any ML model to provide prediction regions with marginal coverage guarantee. The quality of intervals depends on the performance of the underlying ML model and the choice of non-conformity score. Our intention is to emphasize that when the underling ML model is overfitting, our Post-StoNet method can mitigate the overfitting issue and provide shorter intervals with the correct coverage rate. In the revision, we will add another baseline Comformalized Quantile Regression[1] and vanilla StoNet without penalty. Please refer to our rebuttal to reviewer HASS(R2C3-1) for the results.
>
> > **R3Q5** In Table 1 and 2 of Liang et al. (2022), results of massive datasets and methods are listed, which is not discussed in this paper.
>
> The work by Liang et al. (2022)[2] discussed the application of StoNet in nonlinear sufficient dimension reduction. However, the major goals of this paper are (i) to develop a sparse StoNet framework, showing that it can serve as a bridge for transferring the theory and methods developed for linear models to deep learning models; (ii) to develop a post-StoNet framework, showing that it provides a general method for faithfully quantifying the prediction uncertainty of large-scale deep learning models. Since the goals of the two papers are very different, we did not work on the same examples as those in Liang et al. (2022)[2].
>
>
>
> ### Reference
>   [1]Romano, Yaniv, Evan Patterson, and Emmanuel Candes. "Conformalized quantile regression." Advances in neural information processing systems 32 (2019).
>
>   [2] Liang, Siqi, Yan Sun, and Faming Liang. "Nonlinear Sufficient Dimension Reduction with a Stochastic Neural Network." Advances in Neural Information Processing Systems 35 (2022): 27360-27373.

---

### Official Review · Reviewer_HASS · 2023-11-01

**Soundness:** 3 good
**Presentation:** 3 good
**Contribution:** 2 fair
**Rating:** 5
**Confidence:** 3

**Summary:**

The paper is based on StoNet, a stochastic version of a deep neural network where in each layer, (Gaussian) noise is injected into the latent predictions. The authors study how sparsity regularization influences results obtained from StoNet and suggest to propagate uncertainty through the network to obtain prediction uncertainty.

**Strengths:**

- [S1] Originality: As far as I can tell, the authors are the first to study sparsity regularization in StoNet
- [S2] Clarity: The paper is well written and easy to read

**Weaknesses:**

- [W1] Soundness:
    + Some assumptions (already in StoNet) seem to be not realistic (see C1)
    + It's unclear, how the algorithm actually enforces sparsity (see C2)
    + The empirical experiments are limited (see C3)
    + Theoretical guarantees for selection and subsequent inference are not perfectly clear (see Q1, Q2)
- [W2] Originality / Novelty: It often does not get clear what the additional contribution is when compared to StoNet; see also first point on experiments in C3
- [W3] Quality / Presentation: quality of some graphics is really bad (Fig 2 and 3 almost not possible to read)


### Comments:

- [C1]: There is no one "true parameter" in deep neural networks given their [overparametrization-induced and hidden symmetries](https://openreview.net/pdf?id=FOSBQuXgAq) and I don't see how Assumption A2 holds in practice. While the authors recognize this, writing "Given nonidentifiability of the neural network model, Assumption A2 has implicitly assumed that each $\theta$ is unique up to the loss-invariant transformations, e.g., reordering the hidden neurons of the same hidden layer and simultaneously changing the signs of some weights and biases", it does not get clear what this restriction of the Assumption implies and does certainly not account for scaling symmetries present in ReLU networks or hidden symmetries as mentioned above.
- [C2]: How is the model optimized with a Lasso penalty given that this penalty is [non-smooth and stochastic variants hence do not yield exact-zero solutions](https://arxiv.org/pdf/2307.03571.pdf)?
- [C3]: the experiments
    + mainly present coverage rates and calibration results and it is unclear how much of this performance comes from StoNet itself (missing ablation study)
    + only contain one simulation study with a fixed setup (restrictive)
    + do not elaborate on the selection quality except for the small simulation study (missing empirical evidence)

**Questions:**

- [Q1] what seems a bit like magic to me: the paper proves consistent structure selection but without any requirements on the feature matrix (writing "almost any training data"). Afaik it requires rather restrictive assumptions [e.g., here](https://arxiv.org/pdf/1603.06177.pdf) even in much simpler cases such as $l_1$-regularized linear models. Maybe I overlooked that in all the assumptions in the Appendix. Would be great if authors could elloborate on this.
- [Q2] further: how comes that it requires special techniques -- again even in the linear Lasso model -- to obtain valid inference after selection (post-selection inference) and this is not a problem in this much more complicated network?
- [Q3] See C2

---

> ### Author Response · Authors · 2023-11-20
> **Rebuttal by Authors**
>
> Thank you for your constructive reviews and comments, please find our point-by-point response below
>
> > **R2W1-1**  Some assumptions (already in StoNet) seem to be not realistic (see C1)
>
> > **R2C1**   There is no one "true parameter" in deep neural networks given their overparametrization-induced and hidden symmetries and I don't see how Assumption A2 holds in practice. While the authors recognize this, writing "Given nonidentifiability of the neural network model, Assumption A2 has implicitly assumed that each $\theta$
>  is unique up to the loss-invariant transformations, e.g., reordering the hidden neurons of the same hidden layer and simultaneously changing the signs of some weights and biases", it does not get clear what this restriction of the Assumption implies and does certainly not account for scaling symmetries present in ReLU networks or hidden symmetries as mentioned above.
>
> In assumption A2, we essentially group the parameters that define equivalent neural network models into an equivalent class, and the assumption is actually imposed on the equivalent classes of parameters. In practice, the neural network models are usually trained with locally updating algorithms, and it is unlikely to visit all equivalent models. Therefore, practically, assumption A2 is more like a local assumption.
>
> > **R2W1-2** It's unclear, how the algorithm actually enforces sparsity
>
> > **R2C2** How is the model optimized with a Lasso penalty given that this penalty is non-smooth and stochastic variants hence do not yield exact-zero solutions?
>
> In practice, solving a model with LASSO penalty by a gradient descent type algorithm usually employs sub-gradients, where the lasso penalty term gives 0 gradients for the parameters at 0. This use of the sub-gradient in solving Lasso optimization problems has been justified in Nguyen and Yin (2021) [1].
>
> We agree with you that the gradient descent or stochastic gradient descent will not lead to exact 0 parameters. To enforce sparsity in practice, we can put a threshold such as $\lambda$ on the learned parameters. Theoretically, to recover the sparse structure with LASSO penalty,
> one can put a $\beta$-min assumption. For example, in Liu and Yu (2013) [2],
> the authors assumed that
> $$
> n^{\frac{1-c_3}{2}} \min_{1\leq i \leq s }|\beta_i^*| \geq M, \quad \lambda_n \propto n^{\frac{c_4 - 1}{2}},
> $$
> for some constants $M, c_1, c_2, c_3, c_4$ such that $c_1 + c_2 < c_3 \leq 1$, $c_2 < c_4 < c_3 - c_1$. Therefore, using $\lambda$ as a threshold will not miss true connections.
> In practice, similar to Figure 2, with an appropriate choice of $\lambda$, there usually exists a clear threshold to separate true variables and other variables.
>
>
> ### Reference
>
> [1] Nguyen, N.N. and Yin, G. (2021). Stochastic Approximation with Discontinuous Dynamics,
> Differential Inclusions, and Applications.
> arXiv:2108.12652v1.
>
>  [2] Hanzhong Liu and Bin Yu (2013). Asymptotic properties of Lasso+mLS and Lasso+Ridge in sparse high-dimensional linear regression, Electronic Journal of Statistics, 7,
>  3124 - 3169.

---

> ### Author Response · Authors · 2023-11-20
> **Rebuttal by Authors**
>
> > **R2W1-3** The empirical experiments are limited (see C3)
>
> > **R2W2** Originality / Novelty: It often does not get clear what the additional contribution is when compared to StoNet; see also first point on experiments in C3
>
> > **R2C3-1** The experiments mainly present coverage rates and calibration results and it is unclear how much of this performance comes from StoNet itself (missing ablation study)
>
> As one of the main contributions of this paper, we proposed the post-StoNet framework to quantify the prediction uncertainty for large-scale deep learning models, where the StoNet is used as a post hoc calibration model.
> In this framework, we don't need to change the standard DNN training procedure. A StoNet model can be applied on top of the learned representation of the last hidden layer of a "well-trained deep learning model''. This framework is simple while working better than the popular conformal method, especially when the "well-trained deep learning model'' is overfitted. As justified in the paper, the proposed post-StoNet method is able to correctly quantify the prediction uncertainty for the "well-trained deep learning model''.
>
> In the revision, we will also add an ablation study to compare the post-StoNet procedure with and without penalties. In the following table, we can see that for the small data set, a StoNet without penalty failed to provide correct coverage rate, demonstrating the importance of the proposed sparse StoNet structure. And per request by Reviewer GYhy, we also include an additional baseline, Conformalized Quantile Regression(CQR) [1]
>
> | Dataset | N | P | Model | Coverage Rate | Interval length |
> |----------:|:-------:|:--------:|:-------:|:---------:|:--------:|
>  | Wine |  1,599 | 11 | Post-StoNet($\lambda$ = 0.02) | 0.9042(0.0126) |  2.0553(0.0719) |
>   | | | | Post-StoNet($\lambda$ = 0) | 0.2175(0.0209) | 0.2037(0.0304) |
>   | | | | Split Conformal | 0.8958(0.0302) | 2.4534(0.1409) |
>   | | | | Split CQR | 0.9237(0.0132) | 2.3534(0.0501) |
>
> > **R2C3-2** Only contain one simulation study with a fixed setup (restrictive)
>
> > **R2C3-3** do not elaborate on the selection quality except for the small simulation study (missing empirical evidence)
>
> In the rebuttal, we have conducted an additional variable selection
> experiment, which will be added to the paper in the revision. The experiment can be described as follows.
>
> The data are generated from the following high-dimensional nonlinear model ($p=2000$):
> \begin{equation*}
> y = \frac{5x_2}{1 + x_1^2} +5\sin (x_3 x_4) + 2x_5 + 0x_6 + \dots 0x_{2000} + \epsilon
> \end{equation*}
> where $\epsilon \sim N(0,1)$, $x_i = \frac{e + z_i }{\sqrt{2}}$, $e, z_1, \dots, z_{2000} \sim N(0,1)$ are independent. The explanatory $x_i$ are mutually correlated, which makes the problem more challenging. We generate $n=10000$ data for training. We use a StoNet with 1 hidden layer, and 500 hidden units, trained with LASSO penalty $\lambda = 8e-3$. We use $\lambda$ as the threshold to select variables. We repeated the experiment 10 times, the model selects $\\{6,5,5,6,5,5,5,5,5,6\\}$ variables, and all 5 relevant variables are selected.
> Formally, to quantify the selection quality, we can define false selection rate:  $$FSR = \frac{\sum_{i=1}^{10} |\hat{S_i} \setminus S| }{\sum_{i=1}^{10}|\hat{S}_i|}, $$
>
> and negative selection rate: $$NSR = \frac{\sum_{i=1}^{10}|S \setminus \hat{S_i}|}{\sum_{i=1}^{10}|S|},$$  where $S$ is the set of true variables and $\hat{S}_i$ is the set of selected variables in the $i$-th data set.
> Then the corresponding false selection rate is 0.056, and the corresponding negative selection rate is 0.
> This example demonstrates that the StoNet model with a LASSO penalty can correctly identify relevant variables for general nonlinear systems.
>
> ### Reference
>   [1]Romano, Yaniv, Evan Patterson, and Emmanuel Candes. "Conformalized quantile regression." Advances in neural information processing systems 32 (2019).

---

> > ### Author Response · Authors · 2023-11-20
> > **Rebuttal by Authors**
> >
> > > **R2W1-4** Theoretical guarantees for selection and subsequent inference are not perfectly clear (see Q1, Q2)
> >
> > > **R2WQ1** What seems a bit like magic to me: the paper proves consistent structure selection but without any requirements on the feature matrix (writing "almost any training data"). Afaik it requires rather restrictive assumptions e.g., here even in much simpler cases such as $l_1$-regularized linear models. Maybe I overlooked that in all the assumptions in the Appendix. Would be great if authors could elaborate on this.
> >
> > Our results indeed rely on some assumptions on the eigenstructure of the data matrix, specifically, the  $m$-sparse minimum and maximum eigenvalues of the covariance matrix of the hidden variables at each hidden layer. Refer to assumption A3 and Lemma A1 (in the appendix) for the detail. "almost any training data" in the paper refers to the probability of the observed training data set $D_n$ from the data distribution. It doesn't mean that we don't need assumptions on the data distribution.
> >
> > >  **R2Q2** Further: how comes that it requires special techniques -- again even in the linear Lasso model -- to obtain valid inference after selection (post-selection inference) and this is not a problem in this much more complicated network?
> >
> > This is a thoughtful question. If we aim to make inference for the model parameters, a post-selection inference approach is indeed needed. However, in this paper, we aim to make inference for the uncertainty of the prediction. By Theorem 3 of Liu and Yu (2013) [1](select the model then refit selected parameters), the standard post-selection inference procedure (for the model parameters) is not needed, and the proposed inference procedure is valid. We can refit the parameter for the selected structure This point will be mentioned in the revision.
> >
> > > **R2W3** Quality / Presentation: quality of some graphics is really bad (Fig 2 and 3 almost not possible to read)
> >
> > We will add legends for Figures 2 and 3 in the revision. In Figure 2, we will change to 2 colors for relevant variables(5 lines that are far away from $y = 0$ line in the figure) and noise variables.
> >
> >
> > ### Reference
> > [1] Liu, Hanzhong, and Bin Yu. "Asymptotic properties of Lasso+ mLS and Lasso+ Ridge in sparse high-dimensional linear regression." (2013): 3124-3169.

---

> ### Comment · Reviewer_HASS · 2023-11-21
> **Response to Authors' Response**
>
> Thanks for your detailed answer, clarifications and additional experiments. A couple of remaining questions (in case time allows to answer):
>
> > In assumption A2, we essentially group the parameters that define equivalent neural network models into an equivalent class, and the assumption is actually imposed on the equivalent classes of parameters.
>
> How exactly are these equivalent classes defined? Because as far as I understand, symmetries can also be data-dependent (https://arxiv.org/pdf/2210.17216.pdf), exist after accounting for permutation and scaling symmetries (e.g., for hidden symmetries https://arxiv.org/abs/2306.06179), and even if all of these could be accounted for, network weight and function distributions apparently remain multimodal in nature (see, e.g., https://arxiv.org/pdf/2304.02902.pdf, E.2 / why would methods like deep ensemble work so much better than local methods otherwise).
>
> As you mention that your method is "local", this also begs the question what a comparison with local methods like Laplace approximation (https://arxiv.org/abs/2106.14806) and non-local methods like deep ensembles (https://arxiv.org/abs/1612.01474) would look like.
>
> > To enforce sparsity in practice, we can put a threshold such as …
>
> Is it actually possible to simply transfer the arguments from the Liu and Yu paper to your much more complicated setup? A lot of its theory is heavily based on the linear model structure.
>
> And in the first place: Can you actually be sure that optimizing a non-smooth penalty with an optimizer not aware of the non-smoothness will do something meaningful? I thought most of the guarantees require a certain smoothness notion of the underlying function.
>
> > In the following table, we can see that for the small data set …
>
> Just as a side note: I think the Wine dataset is ordinal and would require something like a proportional odds model.

---

> > ### Author Response · Authors · 2023-11-21
> >
> > > **R2R2** Is it actually possible to simply transfer the arguments from the Liu and Yu paper to your much more complicated setup? A lot of its theory is heavily based on the linear model structure.
> > And in the first place: Can you actually be sure that optimizing a non-smooth penalty with an optimizer not aware of the non-smoothness will do something meaningful? I thought most of the guarantees require a certain smoothness notion of the underlying function.
> >
> > Thank you for your valuable feedback. To address this question, let's clarify the structure of our theory.
> >
> > First, the proof for Theorem 1, i.e., Lasso theory transfer from linear models to StoNet, is based on the theoretical framework provided by the imputation-regularized optimization (IRO)
> > algorithm [1](Algorithm 1 in the appendix, it is a stochastic EM type algorithm).
> > In the proof, the regularized regression problem is assumed to be exactly
> > solved at each iteration.
> > Given the imputed value for the hidden units, it corresponds to solving a series of linear regression with LASSO penalty,
> > which is actually doable.
> > Next, we pointed out that the objective function involved in Theorem 1 can also be solved using an adaptive stochastic gradient MCMC algorithm.
> > Therefore, the proof for Theorem 1 is valid based on the IRO algorithm, which does not involve any issue
> > on optimizing a non-smooth penalty with stochastic gradients.
> >
> > Regarding the issue on optimizing a non-smooth penalty with SGD,  particularly for the Lasso penalty,
> > it has been shown in Nguyen and Yin (2021) [2] that SGD works. Therefore, our use of the adaptive SGMCMC for training the StoNet is valid.
> >
> > > **R2R3** Just as a side note: I think the Wine dataset is ordinal and would require something like a proportional odds model.
> >
> > The output of the wine data set is the quality(integer from 0 to 10), we agree with you that a proportional odds model might be more appropriate. We are following the experiments setting of some other stochastic neural network models e.g. [3] where the output is treated as a continuous variable and a normal regression is fitted.
> >
> > ### Reference
> >
> > [1] Liang, Faming, et al. "An imputation–regularized optimization algorithm for high dimensional missing data problems and beyond." Journal of the Royal Statistical Society Series B: Statistical Methodology 80.5 (2018): 899-926.
> >
> > [2] Nguyen, N.N. and Yin, G. (2021). Stochastic Approximation with Discontinuous Dynamics,
> > Differential Inclusions, and Applications.
> > arXiv:2108.12652v1.
> >
> > [3] Hernández-Lobato, José Miguel, and Ryan Adams. "Probabilistic backpropagation for scalable learning of bayesian neural networks." International conference on machine learning. PMLR, 2015.

---

> ### Author Response · Authors · 2023-11-21
>
> Thank you for your valuable feedback, please find our response to your questions below
>
> > **R2R1** How exactly are these equivalent classes defined? Because as far as I understand, symmetries can also be data-dependent (https://arxiv.org/pdf/2210.17216.pdf), exist after accounting for permutation and scaling symmetries (e.g., for hidden symmetries https://arxiv.org/abs/2306.06179), and even if all of these could be accounted for, network weight and function distributions apparently remain multimodal in nature (see, e.g., https://arxiv.org/pdf/2304.02902.pdf, E.2 / why would methods like deep ensemble work so much better than local methods otherwise).
> As you mention that your method is "local", this also begs the question what a comparison with local methods like Laplace approximation (https://arxiv.org/abs/2106.14806) and non-local methods like deep ensembles (https://arxiv.org/abs/1612.01474) would look like.
>
> Thank you for your valuable feedback. Here we would like to further clarify that the equivalent classes are defined implicitly, such that all equivalent models (after appropriate transformations) are mapped to a single model within an appropriately defined set of DNN models.
> We also agree with you that such an equivalent class can be data dependent, and the network weights
> and function distributions remain multimodal in nature.
>
> However, regarding statistical inference for such multimodal models, we want to elaborate by the problem
> of label switching in mixture models (see, for example, [1]). It is almost the same as the problem we are considering for DNNs but simpler. As shown in Liang and Wong (2001)[2], for statistical inference of such a label-switching mixture model, it is enough to conduct inference for one of its component. In particular, Liang and Wong (2001) developed an efficient Monte Carlo algorithm to sample all components of the model and tested the issue of using samples from all components or from a single component. They concluded that statistical inference based on all components of the model is equivalent to that based on a single component.  Based on this result, we state that our theory is developed based on a ``local/single'' component.
>
> Bearing in mind the label-switching mixture model as a prototype of the DNN model, algorithms that make inference based on a local/single model or a mixture of multiple models are all valid, as long as
> the models (optimized or sampled) are consistent. This is exactly the goal of this paper: we aim to
> find a consistent model to validate the downstream inference for deep learning.
> Note that our post-StoNet procedure is to find a consistent model for the data transformed via a well-trained DNN.
>
> Finally, we note that in terms of inference, the Laplace approximation method needs to approximate the Hessian matrix, which will be hard to do for large network models. On the other hand, deep ensemble model[3] lacks theoretical guarantee for its consistency in model estimation, and the resulting inference might not be reliable.
>
> ### Reference
>
> [1] Stephens, Matthew. "Dealing with label switching in mixture models." Journal of the Royal Statistical Society: Series B (Statistical Methodology) 62.4 (2000): 795-809.
>
> [2] Liang, Faming, and Wing Hung Wong. "Real-parameter evolutionary Monte Carlo with applications to Bayesian mixture models." Journal of the American Statistical Association 96.454 (2001): 653-666.
>
> [3] Lakshminarayanan, Balaji, Alexander Pritzel, and Charles Blundell. "Simple and scalable predictive uncertainty estimation using deep ensembles." Advances in neural information processing systems 30 (2017).

---

### Official Review · Reviewer_9U3v · 2023-11-05

**Soundness:** 3 good
**Presentation:** 2 fair
**Contribution:** 2 fair
**Rating:** 5
**Confidence:** 2

**Summary:**

The authors present a follow up on the work on StoNet, a model where the intermediate outputs of the layer are treated as latent variables.

The authors provide several results helping to understand the behaviour of StoNet and empirical simulations showing its performance on real-world data.

**Strengths:**

The paper involves an interesting idea of StoNet, but this appears to be heavily based on the previous work.

**Weaknesses:**

The differences in Table 1 look very small and I wonder if they are statistically significant at all?

I think more empirical evidence would make the paper stronger. The authors discuss scalability, but there's only one experiment with bigger networks (Table 2) where the gains are very marginal?

I don't think overparametrisation of the NNs is necessarily a bad thing, it seems more like an open research question?

The proposed MAP learning does not really utilise the power of the probabilistic model.

The authors should compare in more detail to existing approaches introducing noise to the network, e.g. Gaussian dropout. Overall, I think the work should be more linked to the existing research.

There are works considering similar treatment as section 4 that should be at least cited, e.g. [1].

The developed theory mostly relies on the convergence of MAP/MLE, which happens to be very slow in practice.

[1] Anqi Wu, Sebastian Nowozin, Edward Meeds, Richard E. Turner, José Miguel Hernández-Lobato, Alexander L. Gaunt Deterministic Variational Inference for Robust Bayesian Neural Networks

**Questions:**

See weaknesses.

---

> ### Author Response · Authors · 2023-11-20
> **Rebuttal by Authors**
>
> Thank you for your constructive reviews and comments, please find our point-by-point response below
>
> > **R1W(eakness)1** The differences in Table 1 look very small and I wonder if they are statistically significant at all?
>
> Table 1 gives the coverage rate of StoNet with different choices of $\sigma$. Please note that through this example, we intend to show the trend that the coverage rate becomes closer to the target level 95\% as the value of $\sigma$ is decreased.
> Therefore, the results with different values of $\sigma$ are not expected to be significantly different. Moreover, the similar coverage rates reported in the table imply that the performance of the StoNet is not very sensitive to the choice of $\sigma$. This is just as what we expected.
>
> > **R1W2** I think more empirical evidence would make the paper stronger. The authors discuss scalability, but there's only one experiment with bigger networks (Table 2) where the gains are very marginal?
>
> We will add an experiment on CIFAR100 in the revision. The following table shows the prediction accuracy, negative log likelihood loss (NLL), and expected calibration error (ECE) obtained by the ResNet110 model for the dataset
> with different posthoc calibration methods. The training setting for the model
> is the same as that used for the CIFAR10 dataset (see the paper for the details) and we use $\lambda = 5e-5$ for the penalty.
> The comparison shows that the proposed Post-StoNet approach provides better ECE, demonstrating improvement in model calibration.
>
> Our method indeed provides significant improvement over the baselines. For example, in Table 2 of the paper, for the ResNet110 model, we reported the mean and standard deviation of each metric. Post-StoNet achieves $0.70\\%(0.14\\%)$ ECE, while temperature scaling achieves $1.22\\%(0.16\\%)$ ECE. A two-sample t-test with pooled variance will have a $p$-value of $3.95e-7$. Hence, the difference is highly significant. In the table of CIFAR100 results below, we also include the $p$-value of the two-sample t-tests comparing the ECE of our methods with other baselines, which again shows our method provides significant improvements.
>
> Data | Network  |. Method | ACC | NLL | ECE | p-value |
> |----------:|:-------------:|:-------------:|:-------------:|:-------------:|:-------------:|:-------------:|
> | CIFAR100 | ResNet 110    |  No Post Calibration | 73.38\%(0.57\%) | 1.3364(0.0344) | 15.11\%(0.34\%) | 1.648e-26 |
> | | |         Matrix Scaling | 60.62\%(0.98\%) | 4.4136(0.1754) | 31.98\%(0.86\%) | 9.455e-27 |
> | | |       Temp. Scaling | 73.38\%(0.58\%) | 0.9923(0.0210) | 4.64\%(0.23\%) | 1.538e-16 |
> | | |  Post-StoNet | 73.42\%(0.47\%) | 1.0317(0.0219) |  1.73\%(0.22\%) | --- |
>
> We also conducted another experiment with ImageNet data, where we used a pre-trained ResNet152 model from PyTorch. The output of the last hidden layer is 2048. We use 20\% of the validation set(10000 images) as the calibration set for posthoc calibration. The size of the calibration set is relatively small, in this case, we found that training a logistic regression with LASSO penalty gives better results, where we use the parameter of the last hidden layer of the pre-trained model to initialize the parameter of the logistic regression and train the parameter with SGD for 500 epochs with learning rate 0.001, momentum 0.9 and batch size 128. This model can be viewed as a StoNet without hidden layers, or introducing shortcut connections in StoNet. As we discussed in section 6.2, as long as the model gives a consistent estimation, it is still valid. The result is given in the following table. Because of the use of the pre-trained model, we only report one value for each metric, the Post-StoNet still makes clear improvement in terms of NLL and ECE compared to the original model.
> Data | Network  |. Method | ACC | NLL | ECE |
> |----------:|:-------------:|:-------------:|:-------------:|:-------------:|:-------------:|
> | ImageNet | ResNet 152    |  No Post Calibration | 82.22\% | 0.8707 | 12.74\% |
> | | |  Post-StoNet | 81.85\% | 0.7734 |  1.81\% |
>
> > **R1W3**  I don't think overparameterization of the NNs is necessarily a bad thing, it seems more like an open research question?
>
> We agree with you that overparameterization is not necessarily bad. In terms of optimization, many works have shown good properties of the energy landscape of over-parameterized models. However,  how to quantify the prediction uncertainty of such an over-parameterized model is largely an open problem.
>
> The split conformal method provides a possible way to address this problem, but the resulting
> prediction confidence interval can be overly wide when the model is over-fitted (typically happens for over-parameterized neural networks).
> We address this problem using the proposed post-StoNet method. It adapts the learned model to be
> consistent in prediction, leading to faithful prediction intervals.

---

> ### Author Response · Authors · 2023-11-20
> **Rebuttal by Authors**
>
> > **R1W4** The proposed MAP learning does not really utilise the power of the probabilistic model.
>
> In section 3.2, we estimate the parameter $\theta$ via solving an integration equation (i.e., the equation appeared at the bottom of page 5) using an adaptive stochastic gradient MCMC algorithm. The resulting estimator can be viewed as a maximum a posteriori (MAP) estimator or a regularized frequentist estimator. The power of the StoNet, which enables
> uncertainty quantification for its prediction, comes from its latent variable structure.
> A fully Bayesian treatment for the parameters of StoNet is not necessary at least for the goal of
> this paper.
>
> > **R1W5** The authors should compare in more detail to existing approaches introducing noise to the network, e.g. Gaussian dropout. Overall, I think the work should be more linked to the existing research.
>
> In the revision, we will expand the 'related work' paragraph to include more discussions on other types of stochastic neural networks (that introduce noise to the network). Specifically, the dropout-type methods primarily focus on reducing overfitting, and many other stochastic neural networks are mostly based on heuristic arguments or frame the problem within a framework of variational inference. In Table 4 of Sun and Liang (2022) [2], a kernel-expanded StoNet (similar to the one studied in the paper) and some other stochastic methods (including dropout, variational inference, probabilistic back-propagation) have been compared in terms of prediction. The numerical results indicate that StoNet generally outperforms the others.
>
> As mentioned in the paper, one of our major goals is to provide a mathematically rigorous approach, post-StoNet, for quantifying the prediction uncertainty of large-scale deep learning models. However, the existing stochastic neural networks are generally invalid for prediction uncertainty quantification. In particular, it is unclear how to make statistical inferences with the dropout method.
>
> > **R1W6** There are works considering similar treatment as section 4 that should be at least cited, e.g.[1]
>
> Thank you for pointing out the reference. It also discussed the approximate distribution of the hidden layers, but it is under the Variational Bayes framework, where the parameter of each layer follows the variational distribution, and the goal is to compute Evidence Lower Bound (ELBO).
> In the revision, we will cite and discuss it
> as a piece of related works.
>
>
> > **R1W7** The developed theory mostly relies on the convergence of MAP/MLE, which happens to be very slow in practice.
>
> The proposed algorithm decomposes
> the nonconvex high-dimensional DNN training problem into a series of low-dimensional convex optimization problems, each corresponding to solving a Lasso linear/logistic regression problem.
> Since the dimension of parameters involved in each of the Lasso optimization problems is low, the convergence of the algorithm should not be a concern.
> Furthermore, in section 6.2, the StoNet model
> we used in the post-StoNet approach is usually small.
> For example, for the CIFAR10 example, the post-StoNet model we used has only one hidden layer with 100 hidden units. Again, the computation for such a small model should not be a concern.
>
>
> ### Reference
>
>  [1] Anqi Wu, Sebastian Nowozin, Edward Meeds, Richard E. Turner, Jos\'e Miguel Hern\'andez-Lobato, Alexander L. Gaunt Deterministic Variational Inference for Robust Bayesian Neural Networks.
>
> [2] Sun, Yan, and Faming Liang. "A kernel-expanded stochastic neural network." Journal of the Royal Statistical Society Series B: Statistical Methodology 84.2 (2022): 547-578.

---

### Meta-Review · Area_Chair_1ACc · 2023-12-05

**Metareview:**

This paper proposes a statistical inference framework for rigorous verification in applications of deep learning. Specifically, this paper considered a network called StoNet and used learning theory for sparse estimation to create a framework for statistical properties and statistical inference. The constructed theory is sound and the experimental coverage probabilities are nice. Statistical inference is often misunderstood in the machine learning community, but the experimental results in this paper seem solid. In contrast, the problem of this paper is that the theory, especially the assumptions imposed, are the same as those used in classical statistics and have not been extended for deep learning. For example, assumption A2 does not hold for singular models such as neural networks. The authors state in the text that it can be interpreted as an equivalence class in the rebuttal, but there is no mathematically sound formulation in the paper. Statistical inference without such classical statistical assumptions is a difficult problem of deep learning, and at least in this paper, it is not achieved. To develop valid statistical inference, it will be necessary to extend and adapt common assumptions in this area.

**Justification For Why Not Higher Score:**

The weaknesses of this paper are clearly identified and there is consensus among the reviewers that there is room for improvement.

**Justification For Why Not Lower Score:**

N/A

---

### Decision · Program_Chairs · 2024-01-16

Reject